# STNet: Spectral Transformation Network for Solving Operator Eigenvalue Problem

**Hong Wang[1,2,3]\***, **Yixuan Jiang[1,4]\***, **Jie Wang[1,2,3]†**, **Xinyi Li[1]**, **Jian Luo[1,2,3]**,
**Huanshuo Dong[1,2,3]**

[1] University of Science and Technology of China
[2] CAS Key Laboratory of Technology in GIPAS, University of Science and Technology of China
[3] MoE Key Laboratory of Brain-inspired Intelligent Perception and Cognition, University of Science and Technology of China
[4] Tsinghua University

wanghong1700@mail.ustc.edu.cn, jiangyix25@mails.tsinghua.edu.cn, jiewangx@ustc.edu.cn

## Abstract

Operator eigenvalue problems play a critical role in various scientific fields and engineering applications, yet numerical methods are hindered by the curse of dimensionality. Recent deep learning methods provide an efficient approach to address this challenge by iteratively updating neural networks. These methods' performance relies heavily on the spectral distribution of the given operator: larger gaps between the operator's eigenvalues will improve precision, thus tailored spectral transformations that leverage the spectral distribution can enhance their performance. Based on this observation, we propose the **S**pectral **T**ransformation **Net**work (**STNet**). During each iteration, STNet uses approximate eigenvalues and eigenfunctions to perform spectral transformations on the original operator, turning it into an equivalent but easier problem. Specifically, we employ deflation projection to exclude the subspace corresponding to already solved eigenfunctions, thereby reducing the search space and avoiding converging to existing eigenfunctions. Additionally, our filter transform magnifies eigenvalues in the desired region and suppresses those outside, further improving performance. Extensive experiments demonstrate that STNet consistently outperforms existing learning-based methods, achieving state-of-the-art performance in accuracy [1].

## 1 Introduction

The operator eigenvalue problem is a prominent focus in many scientific fields [10, 4, 6, 35] and engineering applications [8, 5, 13]. However, traditional numerical methods are constrained by the curse of dimensionality, as the computational complexity increases quadratically or even cubically with the mesh size [43].

A promising alternative is using neural networks to approximate eigenfunctions [36]. These approaches reduce the number of parameters by replacing the matrix representation with a parametric nonlinear representation via neural networks. By designing appropriate loss functions, it updates parameters to approximate the desired operator eigenfunctions. These methods only require sampling specific regions without designing a discretization mesh, significantly reducing the algorithm design cost and unnecessary approximation errors [19]. Moreover, neural networks generally exhibit stronger

---

*Equal contribution.
†Corresponding author.
[1]Our code is available at https://github.com/j1y1x/STNet.

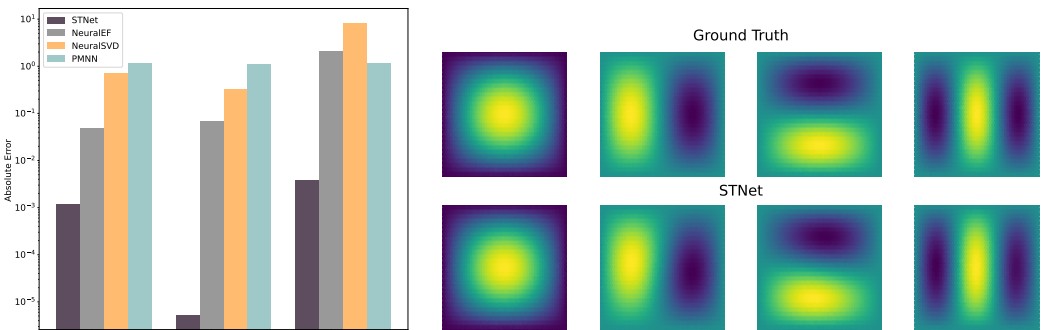

Figure 1: **Left.** Absolute error results of zero eigenvalues for the Fokker-Planck operator computed using various algorithms, the $x$ axis represents the operator dimension. **Right.** Comparison of the eigenfunctions of the 2D Harmonic operator computed by STNet and the ground truth.

expressiveness than linear matrix representations, requiring far fewer sampling points for the same problem compared to traditional methods [34, 14].

Despite these advantages, the performance of such methods strongly depends on the operator's spectral distribution: if the target eigenvalues differs greatly to each other, the algorithm converges much more faster; otherwise, it may suffer from inefficient iterations. To improve convergence, spectral transformations can be designed based on the spectral distribution, reformulating the original problem into an equivalent but more tractable one. However, since the real spectrum of the operator is initially unknown, existing approaches do not optimize spectral properties through such transformations.

To address this limitation, we propose the Spectral Transformation Network (STNet). By exploiting approximate eigenvalues and eigenvectors learned during the iterative process, STNet applies spectral transformations to the original operator, modifying its spectral distribution and thereby converting it into an equivalent problem that converges more easily. Concretely, we employ deflation projection to remove the subspace corresponding to already computed eigenfunctions. This not only narrows the search space but also prevents subsequent eigenfunctions from collapsing into the same subspace. Meanwhile, our filter transform amplifies eigenvalues within the target region and suppresses those outside it, promoting rapid convergence to the desired eigenvalues. Extensive experiments demonstrate that STNet significantly surpasses existing methods based on deep learning, achieving state-of-the-art performance in accuracy. Figure 1 illustrates a selection of the experimental results.

## 2 Preliminaries

### 2.1 Operator Eigenvalue Problem

We primarily focus on the eigenvalue problems of differential operators, such as $\frac{\partial}{\partial x} + \frac{\partial}{\partial y}, \Delta$, etc. Mathematically, an operator $\mathcal{L} : \mathcal{H}_1 \rightarrow \mathcal{H}_2$ is a mapping between two Hilbert spaces. Considering a self-adjoint operator $\mathcal{L}$ defined on a domain $\Omega \subset \mathbb{R}^D$, the operator eigenvalue problem can be expressed in the following form [11]:

$$\mathcal{L}v = \lambda v \quad \text{in } \Omega, \tag{1}$$

where $\Omega \subseteq \mathbb{R}^D$ serves as the domain; $v$ is the eigenfunction and $\lambda$ is the eigenvalue. Typically, it is often necessary to solve for multiple eigenvalues, $\lambda_i, i = 1, \ldots, L$.

### 2.2 Power Method

The power method is a classical algorithm designed to approximate the eigenvalue of an operator $\mathcal{L}$ in the vicinity of a given shift $\sigma$. By applying the shift $\sigma$ (often chosen as an approximation to the target eigenvalue), the original eigenvalue problem is effectively transformed into an equivalent problem for the new operator $(\mathcal{L} - \sigma I)^{-1}$. In each iteration, the current approximate solution is multiplied by this new operator, thereby amplifying the component associated with the eigenvalue closest to $\sigma$. This iterative procedure converges to the desired eigenvalue, as shown in Algorithm 1 [15].

---

**Algorithm 1** Power Method for the Operator $\mathcal{L}$

---

1: **Input:** Operator $\mathcal{L}$, shift $\sigma$, initial guess $v^0$, maximum iterations $k_{\max}$, convergence threshold $\epsilon$.
2: **Output:** Eigenvalue $\lambda$ near $\sigma$.
3: $v^0 = v^0/\|v^0\|$ .
4: **for** $k = 1$ **to** $k_{\max}$ **do**
5:      $v^k = p^k/\|p^k\|$ and solve $(\mathcal{L} - \sigma I)\, p^k = v^{k-1}$.
6:      **if** $\|v^k - v^{k-1}\| < \epsilon$ **then**
7:          $\lambda = \frac{\langle v^k, \mathcal{L} v^k \rangle}{\langle v^k, v^k \rangle}$ and **break**.
8:      **end if**
9: **end for**

---

In each iteration, solving the linear system $(\mathcal{L} - \sigma I)\, p^k = v^{k-1}$ is equivalent to applying the operator $(\mathcal{L} - \sigma I)^{-1}$ to $v^{k-1}$. Afterward, normalizing $v^k$ helps maintain numerical stability. Convergence is typically assessed by evaluating the error $\|v^k - v^{k-1}\|$, ensuring that the final solution meets the desired accuracy. The fundamental reason for the convergence of the power method lies in the repeated application of $(\mathcal{L} - \sigma I)^{-1}$, which progressively magnifies the component of $v^k$ in the direction of the eigenfunction with eigenvalue closest to $\sigma$. For a more detailed introduction to the power method, please refer to the Appendix B.1.

### 2.3 Deflation Projection

The deflation technique plays a critical role in solving eigenvalue problems, particularly when multiple distinct eigenvalues need to be computed. Deflation projection is an effective deflation strategy that utilizes known eigenvalues and corresponding eigenfunctions to modify the structure of the operator, thereby simplifying the computation of remaining eigenvalues [39].

The core idea of deflation projection is to construct an operator $\mathcal{P}$, often defined as $\mathcal{P}(u) = \langle u, v_1 \rangle v_1$ where $v_1$ is a known eigenfunction. This operator is then used to modify the original operator $\mathcal{L}$ into a new operator:

$$\mathcal{B} = \mathcal{L} - \lambda_1 \mathcal{P}. \tag{2}$$

In $\mathcal{B}$, the eigenvalue $\lambda_1$ associated with $v_1$ is effectively removed from the spectrum of $\mathcal{L}$. Additional details on deflation projection can be found in Appendix B.2.

### 2.4 Filter Transform

The filter transform is widely used in numerical linear algebra to enhance the accuracy of eigenvalue computations [39]. By constructing a suitable filter function $F(\mathcal{L})$, the operator $\mathcal{L}$ undergoes a spectral transformation that amplifies the target eigenvalues and suppresses the irrelevant ones. The filter transform can effectively highlight the desired spectral region without altering the corresponding eigenfunctions [43]. Further details on the filter transform can be found in Appendix B.3.

## 3 Method

### 3.1 Problem Formulation

We consider the operator eigenvalue problem for a differential operator $\mathcal{L}$ defined on a domain $\Omega \subset \mathbb{R}^D$. Our goal is to approximate the $L$ eigenvalues $\lambda_i$ near a given shift $\sigma$ and their corresponding eigenfunctions $v_i$, satisfying

$$\mathcal{L}\, v_i = \lambda_i\, v_i, \quad i = 1, 2, \dots, L. \tag{3}$$

To achieve this, we employ $L$ neural networks parameterized by $\theta_i$. Each neural network $NN_{\mathcal{L}}(\cdot; \theta_i)$ maps the domain $\Omega$ into $\mathbb{R}$, providing an approximation of the eigenfunction $v_i$:

$$NN_{\mathcal{L}}(\cdot; \theta_i) : \Omega \to \mathbb{R}, \quad i = 1, 2, \dots, L. \tag{4}$$

In order to represent both the functions and the operators numerically, we discretize $\Omega$ by uniformly randomly sampling $N$ points:

$$S \equiv \{ \boldsymbol{x}_j = (x_j^1, \dots, x_j^D) \mid \boldsymbol{x}_j \in \Omega,\ j = 1, 2, \dots, N \}, \tag{5}$$

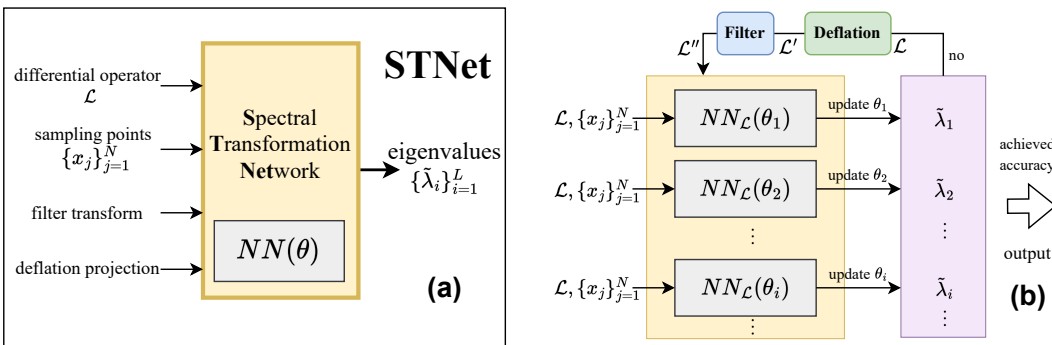

Figure 2: Overview of the **STNet**. **(a)** Introduction to the inputs and outputs. **(b)** STNet comprises multiple neural networks, each tasked with predicting distinct eigenvalues. If the accuracy of the solution reaches the expectation, then STNet will output the result.

Correspondingly, each neural network $NN_{\mathcal{L}}(\cdot; \theta_i)$ output a vector $\boldsymbol{Y}_i \in \mathbb{R}^N$, which approximate the values of the eigenfunction $\tilde{v}_i(\cdot) = NN_{\mathcal{L}}(\cdot; \theta_i)$ at these sampled points:

$$\tilde{v}_i(\boldsymbol{x}_j) \equiv \boldsymbol{Y}_i(j), \quad i = 1, 2, \ldots, L, \quad j = 1, 2, \ldots, N. \tag{6}$$

The approximate eigenvalues $\tilde{\lambda}_i$ are then obtained by applying $\mathcal{L}$ to the computed eigenfunctions $\tilde{v}_i$:

$$\tilde{\lambda}_i \equiv \frac{\langle \tilde{v}_i, \mathcal{L}\tilde{v}_i \rangle}{\langle \tilde{v}_i, \tilde{v}_i \rangle}, \quad i = 1, 2, \ldots, L. \tag{7}$$

Here, the differential operator $\mathcal{L}$ acts on the functions via automatic differentiation. We iteratively update the neural network parameters $\theta_i$ using gradient descent, aiming to minimize the overall residual. Specifically, we formulate the following optimization problem:

$$\min_{\theta_i \in \Theta} \frac{1}{N} \sum_{i=1}^{L} \sum_{j=1}^{N} [\tilde{v}_i(\boldsymbol{x}_j) - v_i(\boldsymbol{x}_j)]^2, \tag{8}$$

where $\Theta$ denotes the parameter space of the neural networks. This approach does not require any training data, as it relies solely on satisfying the differential operator eigenvalue equations over the domain $\Omega$. Finally, this procedure provides approximations $\tilde{\lambda}_i$ of the true eigenvalues $\lambda_i$, $i = 1, \ldots, L$.

### 3.2 Spectral Transformation Network

Inspired by the power method and the power method neural network [45], we propose STNet to solve eigenvalue problems, as shown in Figure 2. In STNet, we replace the function $v^k$ from the $k$-th iteration of the power method with $\tilde{v}_i^k(x) \equiv NN_{\mathcal{L}}(x; \theta_i^k)$, where each neural network is implemented via a multilayer perceptron (MLP). Since neural networks cannot directly perform the inverse operator $(\mathcal{L} - \sigma I)^{-1}$, we enforce $(\mathcal{L} - \sigma I)\tilde{v}^k \approx \tilde{v}^{k-1}$ through a suitable loss function. The updated parameters $\theta_i^k \to \theta_i^{k+1}$ then yield $\tilde{v}^{k+1} = NN_{\mathcal{L}}(x; \theta_i^{k+1})$. Algorithm 2 shows the detailed procedure of STNet.

Classical power method convergence is strongly influenced by the spectral distribution of the operator, which is unknown initially and thus difficult to optimize against directly. However, as the iterative process starts, we can get additional information—such as already computed eigenvalues and eigenfunctions. Using these results for the spectral transformation of the original operator can greatly improve subsequent power-method iterations. In Algorithm 2, we introduce two modules to improve the performance. Their impact on the operator spectrum is shown in Figure 3.

- **Deflation projection** uses already computed eigenvalues and eigenfunctions to construct a projection that excludes the previously resolved subspace, preventing convergence to known eigenfunctions and reducing the search space.
- **Filter transform** employs approximate eigenvalues to construct a spectral transformation (filter function) that enlarges the target eigenvalue region and suppresses others, boosting the efficiency of STNet.

---

**Algorithm 2** Spectral Transformation Network

---

1: **Input:** Operator $\mathcal{L}$ over domain $\Omega \subset \mathbb{R}^D$, shift $\sigma$, number of sampling points $N$, number of eigenvalues $L$, learning rate $\eta$, convergence threshold $\epsilon$, maximum iterations $k_{\max}$.
2: **Output:** Eigenvalues $\tilde{\lambda}_i, \quad i = 1, \dots, L$.
3: Uniformly randomly sample $N$ points $\{\boldsymbol{x}_j\}$ in $\Omega$ to form dataset $S$.
4: Randomly initialize the network parameters $\theta_i^0$, as well as the normalized $\tilde{v}_i$, and set $\tilde{\lambda}_i = \sigma$, $i = 1, \dots, L$.
5: **for** $k = 1$ to $k_{\max}$ **do**
6:     $\tilde{v}_i^k(\boldsymbol{x}_j) = NN_{\mathcal{L}}(\boldsymbol{x}_j; \theta_i^k), \boldsymbol{x}_j \in S$.
7:     $\mathcal{L}_i' = D_i(\mathcal{L}), \quad i = 1, \dots, L$    **// Deflation Projection**
8:     $\mathcal{L}_i'' = F_i(\mathcal{L}'), \quad i = 1, \dots, L$    **// Filter Transform**
9:     $\tilde{u}_i^k(\boldsymbol{x}_j) = \frac{\mathcal{L}_i'' \tilde{v}_i^k(\boldsymbol{x}_j)}{\left\| \mathcal{L}_i'' \tilde{v}_i^k(\boldsymbol{x}_j) \right\|}, \quad i = 1, \dots, L$.
10:     $\text{Loss}_i^k = \frac{1}{N} \sum_{j=1}^N [\tilde{v}_i^{k-1}(\boldsymbol{x}_j) - \tilde{u}_i^k(\boldsymbol{x}_j)]^2, \quad i = 1, \dots, L$.
11:     $\theta_i^{k+1} = \theta_i^k - \eta \nabla_{\theta_i} \text{Loss}_i^k, \quad i = 1, \dots, L$    **// Parameter Update**
12:     **for** $i = 1$ to $L$ **do**
13:         **if** $\text{Loss}_i^k < \epsilon_i$ **then**
14:             $\epsilon_i = \text{Loss}_i^k, \tilde{\lambda}_i = \frac{\langle \tilde{v}_i^k, \mathcal{L} \tilde{v}_i^k \rangle}{\langle \tilde{v}_i^k, \tilde{v}_i^k \rangle}, \tilde{v}_i = \tilde{v}_i^k$.
15:         **end if**
16:     **end for**
17:     **if** $\epsilon_i < \epsilon$ for all $i$ **then**
18:         Convergence achieved; **break**.
19:     **else**
20:         Update deflation projection and filter function: $D_i, F_i, \quad i = 1, \dots, L$.
21:     **end if**
22: **end for**

---

### 3.2.1 Deflation Projection

Suppose we have already approximated the eigenvalues $\tilde{\lambda}_1, \tilde{\lambda}_2, \dots, \tilde{\lambda}_{i-1}$ and their corresponding eigenfunctions $\tilde{v}_1, \tilde{v}_2, \dots, \tilde{v}_{i-1}$. To compute the $i$-th eigenfunction, we focus on the residual subspace orthogonal to the subspace spanned by these previously computed eigenfunctions. The deflated projection is then defined as

$$D_i(\mathcal{L}) \equiv \mathcal{L} - \mathcal{Q}_{i-1} \Sigma_{i-1} \mathcal{Q}_{i-1}^\top. \tag{9}$$

Here $\mathcal{Q}_{i-1}$ maps each vector $(\alpha_1, \dots, \alpha_{i-1}) \in \mathbb{R}^{i-1}$ to the function $\sum_{k=1}^{i-1} \alpha_k \tilde{v}_k$, thus reconstructing functions from the span of $\{\tilde{v}_1, \dots, \tilde{v}_{i-1}\}$. $\mathcal{Q}_{i-1}^\top$ is the transpose of $\mathcal{Q}_{i-1}$. And $\Sigma_{i-1}$ is a diagonal operator that scales each $\tilde{v}_k$ by its corresponding eigenvalue $\tilde{\lambda}_k$.

By employing the deflation projection, the gradient descent search space of the neural network is constrained to be orthogonal to the subspace spanned by $\{\tilde{v}_1, \tilde{v}_2, \dots, \tilde{v}_{i-1}\}$. This projection prevents the neural network output $NN_{\mathcal{L}}(\theta_i)$ from converging to the invariant subspace formed by known eigenfunctions, thereby enhancing the orthogonality among the outputs of different neural networks $NN_{\mathcal{L}}(\theta_1), \dots, NN_{\mathcal{L}}(\theta_{i-1})$. On the one hand, this reduction in the search space accelerates the convergence toward the eigenfunctions $v_i$; On the other hand, it improves the orthogonality among the neural network outputs, which reduces the error in predicting the eigenfunction $\tilde{v}_i$.

In practice, we use the approximate eigenvalues and eigenfunctions with the smallest error in iterations to construct the deflation projection. This allows us to update adaptively, ensuring that the method remains effective when calculating more eigenfunctions.

### 3.2.2 Filter Transform

During the iterative process, we can obtain approximate eigenvalues $\tilde{\lambda}_i$, and assume the corresponding true eigenvalues lie within $[\tilde{\lambda}_i - \xi, \tilde{\lambda}_i + \xi]$, where $\xi$ is a tunable parameter, typically $\xi = 0.1$ or $\xi = 1$. We employ a rational function-based filter transform on the original operator to simultaneously

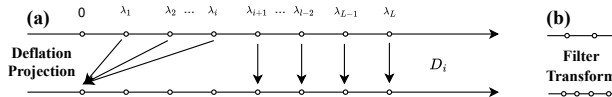 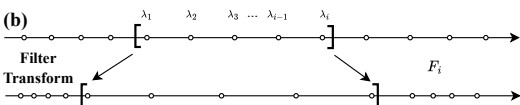

Figure 3: Illustration of the modules' impact on the operator spectrum: **(a)** Deflation projection sets the solved eigenvalues to zero, **(b)** Filter transform enlarges the target eigenvalue region and suppresses others.

amplify the eigenvalues in these intervals and thus improve convergence performance. Specifically, we transform

$$\mathcal{L} \; \longrightarrow \; \prod_{i_0=0}^{i-1} \left[ (\mathcal{L} - (\tilde{\lambda}_{i_0} - \xi)I)\,(\mathcal{L} - (\tilde{\lambda}_{i_0} + \xi)I) \right]^{-1}. \tag{10}$$

By contrast, the basic power method shift-invert strategy, $\mathcal{L} \to (\mathcal{L} - \sigma I)^{-1}$, can be viewed as a special case of this more general construction. In STNet, we simulate the inverse operator via a suitably designed loss function. Therefore, the corresponding pseudocode filter function $F$ removes the inverse, namely:

$$F_i(\mathcal{L}) \; = \; \prod_{i_0=0}^{i-1} \left[ (\mathcal{L} - (\tilde{\lambda}_{i_0} - \xi)I)\,(\mathcal{L} - (\tilde{\lambda}_{i_0} + \xi)I) \right]. \tag{11}$$

When $\lambda_i$ lies within $[\tilde{\lambda}_i - \xi, \tilde{\lambda}_i + \xi]$, the poles $\tilde{\lambda}_i \pm \xi$ make $\|F_i(v_i)\|$ sufficiently large for the corresponding eigenvector $v_i$. This repeated amplification causes that direction to dominate in the subsequent iterations, while eigenvalues outside those intervals are gradually suppressed. Consequently, the method converges more efficiently to the desired eigenvalues.

## 4  Experiments

We conducted comprehensive experiments to evaluate STNet, focusing on:

- Solving multiple eigenvalues in the Harmonic eigenvalue problem.
- Solving the principal eigenvalue in the Schrödinger oscillator equation.
- Solving zero eigenvalues in the Fokker-Planck equation.
- Comparative experiment with traditional algorithms.
- The ablation experiments.

**Baselines**: For these experiments, we selected three learning-based methods for computing operator eigenvalues as our baselines: 1. PMNN [45]; 2. NeuralEF [7]; 3. NeuralSVD [38]. NeuralSVD and NeuralEF were implemented using the publicly available code provided by the authors of NeuralSVD. PMNN was implemented using the code provided by the authors of PMNN. An introduction to related works can be found in Appendix A. In the comparative experiments with traditional algorithms, we chose the finite difference method (FDM) [24].

**Experiment Settings**: To ensure consistency, all experiments were conducted under the same computational conditions. For further details on the experimental environment and algorithm parameters, please refer to Appendices C.1 and C.2.

### 4.1  Harmonic Eigenvalue Problem

Harmonic eigenvalue problems are common in fields such as structural dynamics and acoustics, and can be mathematically expressed as follows [45, 33]:

$$\begin{cases} -\Delta v = \lambda v, & \text{in } \Omega, \\ v = 0, & \text{on } \partial\Omega. \end{cases} \tag{12}$$

Here $\Delta$ denotes the Laplacian operator. We consider the domain $\Omega = [0,1]^D$ where $D$ represents the dimension of the operator, and the boundary conditions are Dirichlet. In this setting, the eigenvalue

Table 1: Relative error comparison for eigenvalues of Harmonic operators. The first row lists the methods, the second row lists eigenvalue indices, and the first column lists the operator dimensions. The most accurate method is in bold.

| Method | NeuralEF | | | | NeuralSVD | | | | STNet | | | |
|---|---|---|---|---|---|---|---|---|---|---|---|---|
| | $\lambda_1$ | $\lambda_2$ | $\lambda_3$ | $\lambda_4$ | $\lambda_1$ | $\lambda_2$ | $\lambda_3$ | $\lambda_4$ | $\lambda_1$ | $\lambda_2$ | $\lambda_3$ | $\lambda_4$ |
| Dim = 1 | 1.98e+0 | 2.53e+0 | 8.89e-1 | 7.81e-1 | 1.72e-2 | 1.83e-2 | **1.88e-2** | 9.38e-1 | **6.38e-11** | **8.61e-3** | 2.13e-2 | **4.05e-1** |
| Dim = 2 | 1.27e+0 | 1.66e+1 | 1.61e+2 | 1.31e+1 | 7.23e-3 | 7.50e-3 | 7.82e-3 | 7.64e-3 | **5.07e-7** | **1.01e-3** | **2.30e-3** | **2.53e-3** |
| Dim = 5 | 3.98e-1 | 7.24e+0 | 1.16e+1 | 1.12e+1 | 2.88e-2 | 3.56e-2 | 3.36e-2 | 3.31e-2 | **4.66e-6** | **1.60e-6** | **1.05e-6** | **2.20e-5** |

problem has analytical solutions, with eigenvalues and corresponding eigenfunctions given by:

$$\lambda_{n_1,\ldots,n_D} = \pi^2 \sum_{k=1}^{D} n_k^2, \quad n_k \in \mathbb{N}^+, \quad u_{n_1,\ldots,n_D}(x_1,\ldots,x_k) = \prod_{k=1}^{D} \sin(n_k \pi x_k). \quad (13)$$

These experiments aim to calculate the smallest four eigenvalues of the Harmonic operator in $1, 2$ and $5$ dimensions. Since the PMNN model only computes the principal eigenvalue and cannot compute multiple eigenvalues simultaneously, it is not considered for comparison.

Firstly, as demonstrated in Table 1, the accuracy of STNet in most tasks is significantly better than that of existing methods. This enhancement primarily stems from the deflation projection. It effectively excludes solved invariant subspaces during the multi-eigenvalue solution process, thereby preserving the accuracy of multiple eigenvalues. This strongly validates the efficacy of our algorithm.

Table 2: Residual comparison for eigenpairs for solving 5-dimensional Harmonic eigenvalue problems. The first row indicates the eigenpair index.

| Index | $(v_1, \lambda_1)$ | $(v_2, \lambda_2)$ | $(v_3, \lambda_3)$ | $(v_4, \lambda_4)$ |
|---|---|---|---|---|
| NeuralSVD | 1.90e+0 | 2.63e+0 | 2.70e+0 | 3.02e+0 |
| NeuralEF | 3.45e+1 | 2.69e+2 | 2.10e+1 | 1.83e+1 |
| STNet | 4.864e-4 | 3.060e-3 | 5.980e-3 | 4.447e-3 |

Secondly, in 5-dimension, STNet consistently maintains a precision improvement of at least three orders of magnitude. As shown in Table 2, this is largely due to the STNet computed eigenpairs having smaller residuals (defined as $||\mathcal{L}\tilde{v} - \tilde{\lambda}\tilde{v}||_2$, see Appendix C.3 for details), indicating that STNet can effectively solve for accurate eigenvalues and eigenfunctions simultaneously.

## 4.2 Schrödinger Oscillator Equation

The Schrödinger oscillator equation is a common problem in quantum mechanics, and its time-independent form is expressed as follows:

$$-\frac{1}{2}\Delta\psi + V\psi = E\psi, \quad \text{in } \Omega = [0,1]^D, \quad (14)$$

where $\psi$ is the wave function, $\Delta$ represents the Laplacian operator indicating the kinetic energy term, $V$ is the potential energy within $\Omega$, and $E$ denotes the energy eigenvalue [38, 16]. This equation is formulated in natural units, simplifying the constants involved. Typically, the po-

Table 3: Relative error comparison for the principal eigenvalues of oscillator operators. The first row lists the methods, and the first column lists the operator dimensions. The most accurate method is in bold.

| Method | PMNN | NeuralEF | NeuralSVD | STNet |
|---|---|---|---|---|
| Dim = 1 | 2.34e-6 | 4.58e+0 | 1.25e-3 | **7.24e-7** |
| Dim = 2 | 9.07e-5 | 3.74e+0 | 5.95e-2 | **2.35e-6** |
| Dim = 5 | 1.57e-1 | 1.78e+0 | 3.32e-1 | **1.29e-1** |

tential $V(x_1,\ldots,x_D) = \frac{1}{2}\sum_{k=1}^{D} x_k^2$ characterizes a multidimensional quadratic potential. The principal eigenvalue $E_0$ and corresponding eigenfunction $\psi_0$ are given by:

$$E_0 = \frac{D}{2}, \quad \psi_0(x_1,\ldots,x_D) = \prod_{k=1}^{D} \left(\frac{1}{\pi}\right)^{\frac{1}{4}} e^{-\frac{x_k^2}{2}}. \quad (15)$$

This experiment focuses on calculating the ground states of the Schrödinger equation in one, two, and five dimensions, i.e., the smallest principal eigenvalues.

Firstly, as shown in Table 3, the STNet achieves significantly higher precision than existing algorithms in computing the principal eigenvalues of the oscillator operator.

Furthermore, the accuracy of STNet surpasses that of PMNN. Both are designed based on the concept of the power method. When solving for the principal eigenvalue, the deflation projection loss may be considered inactive. This outcome suggests that the filter transform significantly enhances the accuracy.

### 4.3 Fokker-Planck Equation

The Fokker-Planck equation is central to statistical mechanics and is extensively applied across diverse fields such as thermodynamics, particle physics, and financial mathematics [45, 21, 12]. It can be mathematically formulated as follows:

$$-\Delta v - V \cdot \nabla v - \Delta V v = \lambda v, \quad \text{in } \Omega = [0, 2\pi]^D, \quad V(x) = \sin\left(\sum_{i=1}^{D} c_i \cos(x_i)\right). \quad (16)$$

Here $V(x)$ is a potential function with each coefficient $c_i$ varying within $[0.1, 1]$, $\lambda$ the eigenvalue, and $v$ the eigenfunction. When the boundary conditions are periodic, there are multiple zero eigenvalues.

The eigenvalue at zero significantly impacts the numerical stability of the algorithm during iterative processes. This experiment investigates the computation of two zero eigenvalues for the Fokker-Planck equations with different parameters in 1, 2, and 5 dimensions. Due to the inherent limitation of the PMNN method, which can only compute a single eigenvalue, we restrict our analysis to calculating one eigenvalue when employing this approach.

Table 4: Absolute error comparison for the zero eigenvalues of Fokker-Planck operators across algorithms. The first row lists the methods, the second row lists the eigenvalue index, the first column lists the Fokker-Planck parameter, and the second column lists the operator dimensions. The most accurate method is in bold.

| Method | | PMNN | NeuralEF | | NeuralSVD | | STNet | |
|---|---|---|---|---|---|---|---|---|
| $c_i$ | Dim | $\lambda_1$ | $\lambda_1$ | $\lambda_2$ | $\lambda_1$ | $\lambda_2$ | $\lambda_1$ | $\lambda_2$ |
| | 1 | 1.16e+0 | 4.98e-2 | 1.05e+0 | 7.19e-1 | 1.02e+0 | **1.17e-3** | **8.75e-3** |
| 0.5 | 2 | 1.11e+0 | 6.71e-2 | 1.57e+0 | 3.33e-1 | 1.03e+0 | **5.26e-6** | **5.14e-2** |
| | 5 | 1.17e+0 | 2.11e+0 | 9.17e+0 | 2.11e+0 | 4.82e+0 | **3.90e-3** | **1.29e-1** |
| | 1 | 8.60e-1 | 5.21e-1 | 5.95e-1 | 2.73e-1 | 3.19e-1 | **3.86e-2** | **2.33e-1** |
| 1.0 | 2 | 8.30e-1 | 6.58e-1 | 8.45e-1 | 2.75e-1 | 3.94e-1 | **1.99e-2** | **3.91e-2** |
| | 5 | 7.58e-1 | 7.71e-1 | 1.02e+0 | 2.01e-1 | 3.08e-1 | **5.64e-2** | **2.67e-2** |

As indicated in Table 4, the STNet algorithm significantly outperforms existing methods in computing the zero eigenvalues of the Fokker-Planck operator, effectively solving cases where the eigenvalue is zero. It is mainly due to the filter function, which performs a spectral transformation on the operator, converting the zero eigenvalue into other eigenvalues that are easier to calculate without changing the eigenvector.

### 4.4 Comparative Experiment with Traditional Algorithms

This experiment compares the accuracy of STNet and the traditional finite difference method (FDM) with a central difference scheme under varying grid densities [24]. Both methods aim to compute the four smallest eigenvalues ($\lambda_1$ to $\lambda_4$) of the 5D harmonic operator.

As shown in Table 5, STNet consistently achieves higher accuracy than FDM across all eigenvalues. While FDM's precision improves with increasing grid density, this comes at the cost of exponentially higher memory consumption. For instance, as the grid points increases from $4^5$ to $45^5$, the relative error for $\lambda_1$ decreases from $3.20 \times 10^{-1}$ to $3.82 \times 10^{-3}$, but memory usage grows from 0.0001 GB to 22.9 GB. This demonstrates the inefficiency of FDM in high-dimensional problems, where maintaining accuracy requires an impractical number of grid points.

In contrast, STNet employs uniform random sampling instead of fixed grids, enabling it to achieve superior accuracy with fewer parameters and lower memory requirements. For example, with a grid

Table 5: Relative error comparison for eigenvalues of 5D Harmonic operators. The first column lists the methods, the second column lists grid points, and the third column lists memory consumption (in GB). Columns $\lambda_1$ to $\lambda_4$ list the eigenvalue relative errors.

| Method | Grid Points | Memory (GB) | $\lambda_1$ | $\lambda_2$ | $\lambda_3$ | $\lambda_4$ |
|---|---|---|---|---|---|---|
| | $4^5$ | 1.01e-4 | 3.20e-1 | 1.04e+0 | 1.04e+0 | 1.04e+0 |
| | $7^5$ | 1.86e-3 | 1.26e-1 | 4.15e-1 | 4.15e-1 | 4.15e-1 |
| | $9^5$ | 6.74e-3 | 8.10e-2 | 2.68e-1 | 2.68e-1 | 2.68e-1 |
| FDM | $15^5$ | 9.05e-2 | 3.17e-2 | 1.05e-1 | 1.05e-1 | 1.05e-1 |
| | $25^5$ | 1.19e+0 | 1.20e-2 | 4.00e-2 | 4.00e-2 | 4.00e-2 |
| | $35^5$ | 6.48e+0 | 6.26e-3 | 2.09e-2 | 2.09e-2 | 2.09e-2 |
| | $45^5$ | 2.29e+1 | 3.82e-3 | 1.27e-2 | 1.27e-2 | 1.28e-2 |
| STNet | $9^5$ | 1.14e+0 | 4.62e-5 | 1.59e-5 | 1.04e-5 | 2.32e-4 |

density of $9^5$, STNet achieves relative errors of $2.32 \times 10^{-4}$ for $\lambda_4$, outperforming FDM by orders of magnitude. Its memory usage remains relatively low at $1.14$ GB, highlighting its scalability and efficiency in high-dimensional eigenvalue problems. By leveraging the expressive power of neural networks, STNet effectively approximates eigenfunctions without relying on dense grids. This makes it a promising alternative to traditional numerical methods for solving high-dimensional operators. A more detailed comparison with traditional algorithms is provided in Appendix D.4.

Traditional algorithms and neural network-based methods each excel in different scenarios. In low-dimensional problems, traditional algorithms are faster and can improve accuracy by increasing grid density. However, in high-dimensional problems, the number of required grid points grows exponentially with dimensionality, making grid-based methods impractical. For example, while a 2D problem requires $100^2$ grid points, a 5D problem requires $100^5$ grid points. Neural network-based algorithms, such as STNet, offer an effective solution to these challenges, providing high accuracy without the need for dense grids.

## 4.5 Ablation Experiments

We conducted ablation experiments to validate the effectiveness of the deflation projection and filter transform modules. As shown in Table 6, the results for "w/o F" indicate that removing the filter transform significantly reduces solution accuracy. In the cases of "w/o D" and "w/o F and D," while the residuals remain small, the absolute errors for $\lambda_2$ and $\lambda_3$ are notably larger compared to $\lambda_1$. This suggests that without the deflation projection module, the network converges exclusively to the first eigenfunction $v_1$ corresponding to $\lambda_1$, failing to capture subsequent eigenfunctions. These findings underscore the critical roles of both modules: the filter transform enhances accuracy through spectral transformation. The deflation projection removes the subspace of already solved eigenfunctions from the search space, enabling the computation of multiple eigenvalues.

Additionally, experiments detailing the performance of STNet as a function of model depth, model width, and the number of sampling points are provided in Appendix D.1. Runtimes and convergence processes for selected experiments are presented in Appendices D.2 and D.3.

Table 6: A comparison of different settings of STNet for the 2-dimensional Harmonic eigenvalue problem. "w/o" denotes the absence of a specific module, "F" represents the filter transform module, and "D" indicates the deflation projection module.

| | Index | $\lambda$ Absolute Error | Residual |
|---|---|---|---|
| | $(v_1, \lambda_1)$ | 1.02e-5 | 4.12e-3 |
| STNet | $(v_2, \lambda_2)$ | 3.04e-2 | 1.24e+1 |
| | $(v_3, \lambda_3)$ | 6.76e-1 | 1.43e+1 |
| | $(v_1, \lambda_1)$ | 6.73e-5 | 1.35e-2 |
| w/o F | $(v_2, \lambda_2)$ | 5.10e-2 | 4.72e+1 |
| | $(v_3, \lambda_3)$ | 1.06e-1 | 1.70e+2 |
| | $(v_1, \lambda_1)$ | 1.42e-5 | 4.12e-3 |
| w/o D | $(v_2, \lambda_2)$ | 2.96e+1 | 7.09e-3 |
| | $(v_3, \lambda_3)$ | 2.97e+1 | 1.09e-2 |
| | $(v_1, \lambda_1)$ | 6.73e-5 | 1.35e-2 |
| w/o F and D | $(v_2, \lambda_2)$ | 2.96e+1 | 1.45e-2 |
| | $(v_3, \lambda_3)$ | 2.97e+1 | 1.37e-2 |

# 5 Limitations and Conclusions

In this paper, we propose STNet, a learning-based approach for solving operator eigenvalue problems. By leveraging approximate eigenvalues and eigenvectors from iterative processes, STNet uses spectral transformations to reformulate the operator, enhancing convergence properties. Experiments show that STNet outperforms existing deep learning methods, achieving state-of-the-art accuracy.

While STNet shows strong performance in solving operator eigenvalue problems, several limitations and avenues for future exploration remain: 1. Although STNet utilizes spectral transformations, the potential benefits of broader matrix preconditioning techniques have not been investigated. 2. Its current scope is limited to linear operators, with future work needed to address nonlinear eigenvalue problems.

## Acknowledgements

The authors would like to thank all the anonymous reviewers for their insightful comments and valuable suggestions. This work was supported by the National Key R&D Program of China under contract 2022ZD0119801, and the National Nature Science Foundations of China grants U23A20388, 62021001.

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

# A Related work

Recent advancements in applying neural networks to eigenvalue problems have shown promising results [42, 9, 30, 26, 27, 28, 37, 23, 29, 25, 18, 31]. Innovations such as spectral inference networks (SpIN) [36], which model eigenvalue problems as kernel problem optimizations solved via neural networks. Neural eigenfunctions (NeuralEF) [7], which significantly reduce computational costs by optimizing the costly orthogonalization steps, are noteworthy. Neural singular value decomposition (NeuralSVD) employs truncated singular value decomposition for low-rank approximation to enhance the orthogonality required in learning functions [38].

Another class of algorithms originates from optimizing the Rayleigh quotient. The deep Ritz method (DRM) utilizes the Rayleigh quotient for computing the smallest eigenvalues, demonstrating significant potential [47]. Several studies have employed the Rayleigh quotient to construct variation-free functions, achieved through physics-informed neural networks (PINNs) [2, 3]. Extensions of this approach include enhanced loss functions with regularization terms to improve the learning accuracy of the smallest eigenvalues [20]. Additionally, [17] reformulates the eigenvalue problem as a fixed-point problem of the semigroup flow induced by the operator, solving it using the diffusion Monte Carlo method. The power method neural network (PMNN) integrates the power method with PINNs, using an iterative process to approximate the exact eigenvalues [45] closely. While PMNN has proven effective in solving for a single eigenvalue [45], it has yet to be developed for computing multiple distinct eigenvalues simultaneously.

# B Background Knowledge and Relevant Analysis

## B.1 Convergence Analysis of the Power Method

Suppose $\boldsymbol{A} \in \mathbb{R}^{n \times n}$ and $\boldsymbol{V}^{-1} \boldsymbol{A} \boldsymbol{V} = \mathrm{diag}(\lambda_1, \ldots, \lambda_n)$ with $\boldsymbol{V} = [\boldsymbol{v}_1 \quad \cdots \quad \boldsymbol{v}_n]$. Assume that $|\lambda_1| > |\lambda_2| \geq \cdots \geq |\lambda_n|$. The pseudocode for the power method is shown below [15]:

---

**Algorithm 1:** Power method for finding the largest principal eigenvalue of the matrix $A$

---

1  **Given** $\boldsymbol{A} \in \mathbb{R}^{n \times n}$ an $n \times n$ matrix, an arbitrary unit vector $x^{(0)} \in \mathbb{R}^n$, the maximum number of iterations $k_{\max}$, and the stopping criterion $\epsilon$.
2  **for** $k = 1, 2, \ldots, k_{max}$ **do**
3  $\quad$ Compute $\boldsymbol{y}^{(k)} = \boldsymbol{A} \boldsymbol{x}^{(k-1)}$.
4  $\quad$ Normalize $\boldsymbol{x}^{(k)} = \frac{\boldsymbol{y}^{(k)}}{\|\boldsymbol{y}^{(k)}\|}$.
5  $\quad$ Compute the difference $\delta = \|\boldsymbol{x}^{(k)} - \boldsymbol{x}^{(k-1)}\|$.
6  $\quad$ **if** $\delta < \epsilon$ **then**
7  $\quad\quad$ Record the largest principal eigenvalue using the Rayleigh quotient.

$$\lambda^{(k)} = \frac{\langle \boldsymbol{x}^{(k)}, \boldsymbol{A} \boldsymbol{x}^{(k)} \rangle}{\langle \boldsymbol{x}^{(k)}, \boldsymbol{x}^{(k)} \rangle}.$$

$\quad\quad$ The stopping criterion is met, and the iteration can be stopped.

---

Let us examine the convergence properties of the power iteration. If

$$\boldsymbol{x}^{(0)} = a_1 \boldsymbol{v}_1 + a_2 \boldsymbol{v}_2 + \cdots + a_n \boldsymbol{v}_n$$

and $\boldsymbol{v}_1 \neq 0$, then

$$\boldsymbol{A}^k \boldsymbol{x}^{(0)} = a_1 \lambda_1^k \left( \boldsymbol{v}_1 + \sum_{j=2}^{n} \frac{a_j}{a_1} \left( \frac{\lambda_j}{\lambda_1} \right)^k \boldsymbol{v}_j \right).$$

Since $\boldsymbol{x}^{(k)} \in \mathrm{span}\{\boldsymbol{A}^k \boldsymbol{x}^{(0)}\}$, we conclude that

$$\mathrm{dist}\left( \mathrm{span}\{\boldsymbol{x}^{(k)}\}, \mathrm{span}\{\boldsymbol{v}_1\} \right) = O\left( \left( \frac{\lambda_2}{\lambda_1} \right)^k \right).$$

It is also easy to verify that

$$|\lambda_1 - \lambda^{(k)}| = O\left(\left(\frac{\lambda_2}{\lambda_1}\right)^k\right).$$

Since $\lambda_1$ is larger than all the other eigenvalues in modulus, it is referred to as the largest principal eigenvalue. Thus, the power method converges if $\lambda_1$ is the largest principal and if $x^{(0)}$ has a component in the direction of the corresponding dominant eigenvector $x_1$.

In practice, the effectiveness of the power method largely depends on the ratio $|\lambda_2|/|\lambda_1|$, as this ratio determines the convergence rate. Therefore, applying specific spectral transformations to the matrix to increase this ratio can significantly accelerate the convergence of the power method.

## B.2 Deflation Projection Details

Consider the scenario where we have determined the largest modulus eigenvalue, $\lambda_1$, and its corresponding eigenvector, $v_1$, utilizing an algorithm such as the power method. These algorithms consistently identify the eigenvalue of the largest modulus from the given matrix along with an associated eigenvector. We ensure that the vector $v_1$ is normalized such that $\|v_1\|_2 = 1$. The task then becomes computing the subsequent eigenvalue, $\lambda_2$, of the matrix $A$. A traditional approach to address this is through what is commonly known as a deflation procedure. This technique involves a rank-one modification to the original matrix, aimed at shifting the eigenvalue $\lambda_1$ while preserving all other eigenvalues intact. The modification is designed in such a way that $\lambda_2$ emerges as the eigenvalue with the largest modulus in the adjusted matrix. Consequently, the power method can be reapplied to this updated matrix to extract the eigenvalue-eigenvector pair $\lambda_2, v_2$.

When the invariant subspace requiring deflation is one-dimensional, consider the following Proposition B.1. The propositions and proofs below are derived from [39] P90.

**Proposition B.1.** *Let $v_1$ be an eigenvector of $A$ of norm 1, associated with the eigenvalue $\lambda_1$ and let $A_1 \equiv A - \sigma v_1 v_1^H$. Then the eigenvalues of $A_1$ are $\tilde{\lambda}_1 = \lambda_1 - \sigma$ and $\tilde{\lambda}_j = \lambda_j, j = 2, 3, \ldots, n$. Moreover, the Schur vectors associated with $\tilde{\lambda}_j, j = 1, 2, 3, \ldots, n$ are identical with those of $A$.*

*Proof.* Let $AV = VR$ be the Schur factorization of $A$, where $R$ is upper triangular and $V$ is orthonormal. Then we have

$$A_1 V = \left[A - \sigma v_1 v_1^\top\right] V = VR - \sigma v_1 e_1^\top = V[R - \sigma e_1 e_1^\top].$$

Here, $e_1$ is the first standard basis vector. The result follows immediately. $\square$

According to Proposition B.1, once the eigenvalue $\lambda_1$ and eigenvector $v_1$ are known, we can define the deflation projection matrix $P_1 = I - \lambda_1 v_1 v_1^\top$ to compute the remaining eigenvalues and eigenvectors.

When deflating with multiple vectors, let $q_1, q_2, \ldots, q_j$ be a set of Schur vectors associated with the eigenvalues $\lambda_1, \lambda_2, \ldots, \lambda_j$. We denote by $Q_j$ the matrix of column vectors $q_1, q_2, \ldots, q_j$. Thus, $Q_j \equiv [q_1, q_2, \ldots, q_j]$ is an orthonormal matrix whose columns form a basis of the eigenspace associated with the eigenvalues $\lambda_1, \lambda_2, \ldots, \lambda_j$. An immediate generalization of Proposition B.1 is the following [39] P94.

**Proposition B.2.** *Let $\Sigma_j$ be the $j \times j$ diagonal matrix $\Sigma_j = diag(\sigma_1, \sigma_2, \ldots, \sigma_j)$, and $Q_j$ an $n \times j$ orthogonal matrix consisting of the Schur vectors of $A$ associated with $\lambda_1, \ldots, \lambda_j$. Then the eigenvalues of the matrix*

$$A_j \equiv A - Q_j \Sigma_j Q_j^\top,$$

*are $\tilde{\lambda}_i = \lambda_i - \sigma_i$ for $i \leq j$ and $\tilde{\lambda}_i = \lambda_i$ for $i > j$. Moreover, its associated Schur vectors are identical with those of $A$.*

*Proof.* Let $AU = UR$ be the Schur factorization of $A$. We have

$$A_j U = \left[A - Q_j \Sigma_j Q_j^\top\right] U = UR - Q_j \Sigma_j E_j^\top,$$

where $E_j = [e_1, e_2, \ldots, e_j]$. Hence

$$A_j U = U\left[R - E_j \Sigma_j E_j^\top\right]$$

and the result follows. $\square$

According to Proposition B.2, if $\boldsymbol{A}$ is a normal matrix and the eigenvalues $\lambda_1, \ldots, \lambda_j$ along with their corresponding eigenvectors $\boldsymbol{v}_1, \ldots, \boldsymbol{v}_j$ are known, we can construct the deflation projection matrix $\boldsymbol{P}_j = \boldsymbol{I} - \boldsymbol{V}_j \boldsymbol{\Sigma}_j \boldsymbol{V}_j^\top$ to compute the remaining eigenvalues and eigenvectors. Here, $\boldsymbol{\Sigma}_j = \mathrm{diag}(\sigma_1, \sigma_2, \ldots, \sigma_j)$ and $\boldsymbol{V}_j = [\boldsymbol{v}_1, \boldsymbol{v}_2, \ldots, \boldsymbol{v}_j]$.

## B.3 Filtering Technique

The primary objective of filtering techniques is to manipulate the eigenvalue distribution of a matrix through spectral transformations [39]. This enhances specific eigenvalues of interest, facilitating their recognition and computation by iterative solvers. Filter transformation functions, $F(x)$, typically fall into two categories:

1. Polynomial Filters, expressed as $P(x)$, such as the Chebyshev filter [32, 1].
2. Rational Function Filters, often denoted as $P(x)/Q(x)$, such as the shift-invert method [41, 43]. Below, we describe this strategy in detail.

**Shift-Invert Strategy**    The shift-invert strategy applies the transformation $(A - \sigma I)^{-1}$ to the matrix $A$, where $\sigma$ is a scalar approximating a target eigenvalue, termed as shift. This operation transforms each eigenvalue $\lambda$ of $A$ into $\frac{1}{\lambda - \sigma}$, amplifying those eigenvalues close to $\sigma$ in the transformed matrix, making them larger and more distinguishable [43].

For instance, consider the power method, where the convergence rate is primarily governed by the ratio of the matrix's largest modulus eigenvalue to its second largest. Suppose matrix $A$ has three principal eigenvalues: $\lambda_1 = 10$, $\lambda_2 = 3$, and $\lambda_3 = 2$. Our objective is to compute $\lambda_1$, the largest eigenvalue. In the original matrix $A$, the convergence rate of the power method hinges on the spectral gap ratio, defined as:

$$\text{Spectral Gap Ratio} = \frac{\lambda_1}{\lambda_2} \approx 3.33$$

Applying the shift-invert transformation with $\sigma = 9.5$ strategically selected close to $\lambda_1$, the new eigenvalues $\mu$ are recalculated as:

$$\mu_i = \frac{1}{\lambda_i - \sigma}$$

This results in transformed eigenvalues:

$$\mu_1 = 2, \quad \mu_2 \approx -0.133, \quad \mu_3 \approx -0.125$$

Under this transformation, $\mu_1 = 2$ emerges as the dominant eigenvalue in the new matrix, with the other eigenvalues significantly smaller. Consequently, the new spectral gap ratio escalates to:

$$\text{New Spectral Gap Ratio} = \frac{2}{0.133} \approx 15.04$$

This enhanced spectral gap notably accelerates the convergence of the power method in the new matrix configuration.

Filtering techniques are often synergized with techniques like the implicit restarts of Krylov algorithms [43, 15, 46], employing matrix operation optimizations to minimize the computational demands of evaluating matrix functions. These methods enable more precise localization and computation of multiple eigenvalues spread across the spectral range, particularly vital in physical [40, 1] and materials science [22] simulations where these eigenvalues frequently correlate with the system's fundamental properties [44].

# C   Details of Experimental Setup

## C.1   Experimental Environment

To ensure consistency in our evaluations, all comparative experiments were conducted under uniform computing environments. Specifically, the environments used are detailed as follows:

- CPU: 72 vCPU AMD EPYC 9754 128-Core Processor
- GPU: NVIDIA GeForce RTX 4090D (24GB)

## C.2 Experimental Parameters

NeuralSVD and NeuralEF were implemented using the publicly available code provided by the authors of NeuralSVD (`https://github.com/jongharyu/neural-svd`). PMNN was implemented using the code provided by the authors of PMNN (`https://github.com/SummerLoveRain/PMNN_IPMNN`). No modifications were made to their original code for the experiments, except for changes to the target operator to be solved. All experiments are fully reproducible. The detailed baseline experimental parameter files and code are provided in the supplementary materials.

- NeuralSVD and NeuralEF: (Using the original paper settings)
    - Optimizer: RMSProp with a learning rate scheduler.
    - Learning rate: 1e-4, batch size: 128
    - Neural Network Architecture: layers = [128,128,128]
    - Laplacian regularization set to 0.01, with evaluation frequency every 10000 iterations.
    - Fourier feature mapping enabled with a size of 256.
    - Neural network structure: hidden layers of 128,128,128 using softplus as the activation function.
- PMNN and STNet
    - Optimizer: Adam
    - Learning rate: 1e-4
    - Neural Network Architecture: Assuming d is the dimension of the operator. For d = 1 or 2, layers = [d, 20, 20, 20, 20, 1]. For d=5, layers = [d, 40, 40, 40, 40, 1]. For the else cases, layers = [d, 40, 40, 40, 40, 1].
    - For the 1-dimensional problem, the number of points is $20,000$, with $400,000$ iterations. For the 2-dimensional problem, the number of points is $40,000 = 200 \times 200$, also with $400,000$ iterations. For the 5-dimensional problem, the number of points is $59,049 = 9^5$, with $500,000$ iterations.

## C.3 Error Metrics

- Absolute Error:
  We employ absolute error to estimate the bias of the output eigenvalues of the model:

$$\text{Absolute Error} = |\tilde{\lambda} - \lambda|.$$

  Here $\tilde{\lambda}$ represents the eigenvalue predicted by the model, while $\lambda$ denotes the true eigenvalue.

- Relative Error:
  We employ relative error to estimate the bias of the output eigenvalues of the model:

$$\text{Relative Error} = \frac{|\tilde{\lambda} - \lambda|}{|\lambda|}.$$

  Here $\tilde{\lambda}$ represents the eigenvalue predicted by the model, while $\lambda$ denotes the true eigenvalue.

- Residual Error:
  To further analyze the error in eigenpair $(\tilde{v}, \tilde{\lambda})$ predictions, we use the following metric:

$$\text{Residual Error} = ||\mathcal{L}\tilde{v} - \tilde{\lambda}\tilde{v}||_2.$$

  Here, $\tilde{v}$ represents the eigenfunction predicted by the model. When $\tilde{\lambda}$ is the true eigenvalue and $\tilde{v}$ is the true eigenfunction, the Residual Error equals 0.

# D   Supplementary Experiments

## D.1   Analysis of Hyperparameters

**Model Depth**:

Table 7: Consider the 2-dimensional Harmonic problem, with the fixed layer width of 20, and compare the performance of STNet at different model layers. Other experimental details are the same as Appendix C.2. The error metric is absolute error.

| Eigenvalue | Layer 3 | Layer 4 | Layer 5 | Layer 6 |
|---|---|---|---|---|
| $\lambda_1$ | 1.02e-5 | 1.42e-5 | 4.36e-6 | 1.06e-5 |
| $\lambda_2$ | 3.04e-2 | 2.96e-1 | 8.63e-1 | 8.21e-1 |
| $\lambda_3$ | 6.76e-2 | 4.17e-1 | 1.98e+0 | 1.17e+0 |
| $\lambda_4$ | 1.00e-1 | 2.00e+1 | 8.94e+1 | 3.81e+1 |

**Model Width**:

Table 8: Consider the 2-dimensional Harmonic problem, with the fixed layer depth of 3, and compare the performance of STNet at different model widths. Other experimental details are the same as Appendix C.2. The error metric is absolute error.

| Eigenvalue | Width 10 | Width 20 | Width 30 | Width 40 |
|---|---|---|---|---|
| $\lambda_1$ | 1.68e-6 | 1.42e-5 | 3.26e-5 | 1.57e-5 |
| $\lambda_2$ | 3.82e-1 | 2.96e-1 | 1.50e+0 | 2.67e+0 |
| $\lambda_3$ | 7.54e-1 | 4.17e-1 | 1.59e+0 | 7.93e+1 |
| $\lambda_4$ | 1.71e-1 | 2.00e+1 | 3.52e+2 | 1.50e+2 |

**The Number of Points**:

Table 9: Consider the 2-dimensional Harmonic problem and compare the performance of STNet at different numbers of points. Other experimental details are the same as Appendix C.2. The error metric is absolute error.

| Eigenvalue | 20000 Points | 30000 Points | 40000 Points | 50000 Points |
|---|---|---|---|---|
| $\lambda_1$ | 1.11e-5 | 4.40e-5 | 1.42e-5 | 4.94e-6 |
| $\lambda_2$ | 1.25e+0 | 3.58e-1 | 2.96e-1 | 2.53e-1 |
| $\lambda_3$ | 1.61e+0 | 1.70e-1 | 4.17e-1 | 3.73e-1 |

The influence of model depth, model width, and the number of points on STNet is illustrated in Tables 7, 8, and 9, respectively. Experimental results indicate that STNet is relatively unaffected by changes in model depth and model width. However, it is significantly influenced by the number of points, with performance improving as more points are used.

## D.2 Computational Complexity and Runtimes

While neural network-based methods may exhibit a larger computational overhead than traditional methods in low-dimensional problems, their strength lies in high-dimensional settings where they offer a significant precision and computational cost advantages. To illustrate this, we provide a runtime comparison for the 5D Harmonic problem in Table 10.

Table 10: Runtime comparison for the 5D Harmonic problem. "STNet (partial)" refers to the time taken to achieve an error of $1.00 \times 10^{-3}$.

| Method | Computation Time (s) | Principal Eigenvalue Absolute Error |
|---|---|---|
| FDM ($45^5$ points) | $\sim 1.9e+4$ | 3.8e-3 |
| STNet (partial) | $\sim 1.3e+3$ | 1.0e-3 |
| STNet (converged) | $\sim 3.1e+4$ | 4.6e-5 |
| NeuralEF (converged) | $\sim 1.2e+4$ | 3.9e-1 |
| NeuralSVD (converged) | $\sim 1.2e+4$ | 2.8e-2 |

The results in Table 10 highlight two key points:

1. STNet achieves a lower error more rapidly than other learning-based methods and converges to a significantly more accurate solution than NeuralEF and NeuralSVD.

2. The total time to full convergence for STNet is longer. This is an inherent trade-off of the Filter Transform, which iteratively refines the problem to unlock further optimization, thus requiring more iterations to reach its high-precision potential.

## D.3 Partial Convergence Process

we provide the convergence process for the Harmonic problem in Table 11. The table shows the absolute error for the first two eigenvalues ($\lambda_1, \lambda_2$) at different iteration counts across dimensions 1, 2, and 5.

Table 11: Absolute Error vs. Iterations for the Harmonic problem. Empty cells indicate that convergence for that eigenvalue was already achieved in a prior stage.

| Dim | Eigenvalue | Iter: 100 | 500 | 1000 | 40000 | 80000 | 130000 | 180000 |
|---|---|---|---|---|---|---|---|---|
| d=1 | $\lambda_1$ | 1.6e-2 | 3.5e-3 | 3.2e-4 | 2.6e-4 | 2.7e-5 | 1.2e-3 | |
| | $\lambda_2$ | 1.0e+4 | 2.4e+3 | 2.8e+3 | 8.0e+2 | 9.8e-1 | 8.8e-3 | |
| d=2 | $\lambda_1$ | 6.9e-2 | 6.9e-3 | 5.9e-3 | 5.3e-6 | | | |
| | $\lambda_2$ | 6.3e01 | 5.9e+0 | 4.2e+0 | 5.1e-2 | | | |
| d=5 | $\lambda_1$ | 5.8e-2 | 6.6e-3 | 4.8e-4 | 7.8e-3 | 3.9e-3 | 6.2e-3 | 3.9e-3 |
| | $\lambda_2$ | 6.3e+3 | 4.1e+2 | 1.5e+2 | 2.9e+0 | 5.5e+0 | 3.8e+0 | 1.3e-1 |

## D.4 Impact of Discretization on Traditional Methods

A critical point of comparison is understanding the primary sources of error in different methodologies. For traditional methods like the Finite Difference Method (FDM), the dominant bottleneck in high-dimensional problems is not the algebraic eigenvalue solver's accuracy but the fundamental discretization error introduced when mapping a continuous operator onto a discrete grid. While advanced solver techniques such as deflation or filtering can improve the precision of the matrix eigenvalue solution, they cannot overcome the inherent error from the initial discretization. Reducing this error necessitates a denser grid, which leads to an exponential increase in memory and computational cost, a challenge known as the "curse of dimensionality."

To empirically validate this, we conducted a experiment for the 5D Harmonic problem, comparing STNet against FDM enhanced with Richardson extrapolation—a technique designed to improve

accuracy. The results, presented in Table 12, show that even when FDM is augmented with advanced techniques, it cannot match the accuracy of STNet.

Table 12: Comparison with advanced FDM techniques for the 5D Harmonic problem.

| Method | Grid/Points | Memory (GB) | $\lambda_1$ Error | $\lambda_2$ Error | $\lambda_3$ Error | $\lambda_4$ Error |
|---|---|---|---|---|---|---|
| FDM (Richardson Extrap.) | $25^5$ | 3.4e+0 | 8.6e-3 | 2.9e-2 | 2.9e-2 | 2.9e-2 |
| FDM (with deflation & filtering; Richardson Extrap.) | $25^5$ | 3.4e+0 | 8.6e-3 | 2.9e-2 | 2.9e-2 | 2.9e-2 |
| FDM (Central Difference) | $45^5$ | 2.2e+1 | 3.8e-3 | 1.2e-2 | 1.2e-2 | 1.2e-2 |
| FDM (with deflation & filtering; Central Difference) | $45^5$ | 2.2e+1 | 3.8e-3 | 1.2e-2 | 1.2e-2 | 1.2e-2 |
| STNet | $\mathbf{9^5}$ | **1.1e+0** | **4.6e-5** | **1.6e-5** | **1.0e-5** | **2.3e-4** |

This experiment reinforces that neural network-based methods, which rely on mesh-free random sampling, are inherently better suited for high-dimensional problems, as they effectively bypass the discretization step that limits traditional grid-based approaches.

While traditional methods are often superior for low-dimensional problems, their practicality diminishes rapidly in high dimensions due to the "curse of dimensionality". To provide a more granular analysis, Table 13 details the convergence time versus accuracy for the 5D Harmonic problem. Our traditional baseline uses 'scipy.eigs', which is based on the highly-optimized ARPACK library. The data clearly shows that FDM's accuracy is capped by the grid resolution, and STNet can achieve higher precision. The total time for STNet is longer, but it reaches a level of accuracy that is infeasible for FDM under reasonable memory constraints.

Table 13: Detailed convergence analysis for the 5D Harmonic problem ($\lambda_1$). The table shows the time required to reach specific absolute error thresholds. A dash (-) indicates that the method could not reach the specified error due to limitations imposed by grid resolution.

| Method | Grid/Sample Pts | Memory (GB) | Final Abs. Error | Time to Error < 1e+0 | Time to Error < 1e-1 | Time to Error < 1e-2 | Time to Error < 1e-3 | Total Time |
|---|---|---|---|---|---|---|---|---|
| FDM+eigs | $15^5$ ($\sim$ 8e+5) | 9.1e-2 | 3.2e-2 | 1.3s | 4.0s | 1.1e+1s | - | 1.1e+1s |
| | $25^5$ ($\sim$ 1e+7) | 1.2e+0 | 1.2e-2 | 2.5e+1s | 3.2e+1s | 8.1e+1s | - | 1.1e+2s |
| | $35^5$ ($\sim$ 5e+7) | 6.5e+0 | 6.3e-3 | 1.8e+2s | 2.5e+2s | 4.9e+2s | 1.9e+3s | 1.9e+3s |
| | $45^5$ ($\sim$ 2e+8) | 2.3e+1 | 3.8e-3 | 9.4e+2s | 3.1e+3s | 6.7e+3s | 1.9e+4s | 1.9e+4s |
| STNet | 5.0e+4 | 9.5e-1 | 3.4e-4 | 7.2s | 1.4e+1s | 1.7e+2s | 1.3e+3s | 1.0e+5s |
| | $9^5$ ($\sim$ 6e+4) | 1.1e+0 | 4.6e-5 | 9.6s | 1.9e+1s | 1.5e+2s | 1.3e+3s | 3.1e+4s |
| | 7.0e+4 | 1.3e+0 | 6.5e-5 | 5.7s | 1.1e+1s | 1.4e+2s | 3.0e+2s | 5.4e+4s |
| | 8.0e+4 | 1.5e+0 | 2.5e-5 | 8.9s | 1.6e+1s | 1.6e+2s | 9.8e+2s | 2.9e+4s |

This detailed analysis validates our conclusion that for large-scale, high-dimensional problems, STNet holds a significant advantage in both achievable accuracy and memory efficiency. Its core strength lies in its mesh-free random sampling strategy, which leverages the expressive power of neural networks to approximate eigenfunctions directly, thereby circumventing the need to construct and store a massive matrix.

