# OpenReview forum: "STNet: Spectral Transformation Network for Solving Operator Eigenvalue Problem"
_NeurIPS.cc/2025/Conference — NeurIPS 2025 poster_

### Official Review · Reviewer_btcB · 2025-07-01

**Clarity:** 3
**Significance:** 2
**Originality:** 2
**Rating:** 4
**Confidence:** 4

**Summary:**

The paper proposes STNet (Spectral Transformation Network), a deep learning framework designed to solve operator eigenvalue problems efficiently, particularly in high-dimensional settings. STNet enhances convergence by applying spectral transformations to the operator using approximate eigenvalues and eigenfunctions obtained during training. Specifically, it employs two key methods: a deflation projection that eliminates previously learned eigencomponents to reduce redundancy and a filter transform that amplifies target spectral regions to accelerate convergence. By integrating these modules into an iterative neural network training process, STNet reformulates the original eigenvalue problem into a series of more tractable ones. Experiments across harmonic, Schrödinger, and Fokker-Planck operators demonstrate that STNet outperforms existing learning-based and traditional methods in accuracy.

**Questions:**

1. Can you compare the convergence with the traditional power method under different matrix sizes?
2. Are there any advantages in computational cost and memory usage for large-scale matrices?
3. Can STNet be applied to real-world problems where the operator or domain is not analytically defined or is noisy?

**Ethical Concerns:**

["NO or VERY MINOR ethics concerns only"]

**Final Justification:**

The general applicability of this work remains uncertain, but this work shows great improvement in some domains. The additional data provided in the rebuttal makes it more convincing in the given settings. Thus, I'd like to raise my score.

**Limitations:**

Yes

**Quality:**

3

**Strengths And Weaknesses:**

Strengths:
- This paper is overall well written and easy to follow. The method and results are clearly presented.
- It combines the deflation and filter transforms into a neural-network-based solver for the operator eigenvalue problem, and this method is described clearly.
- The work addresses a challenging and important problem: solving high-dimensional eigenvalue problems, which arise in many scientific and engineering domains. By demonstrating scalability and improved performance over other learning-based baselines, STNet has potential for meaningful impact in both scientific computing and machine learning communities.

Weaknesses:
- The method is primarily evaluated on synthetic benchmark problems. While these are standard in the literature, applying STNet to real-world or more complex physical systems would better demonstrate its broader applicability.
- Lack of comparison with traditional eigenvalue methods, like the convergence and computational cost.
- There is no explicit study on the convergence for different matrix sizes, and how it scales.

---

> ### Author Rebuttal · Authors · 2025-07-31
>
> Dear Reviewer,
>
> Thank you for your detailed and insightful review of our paper. We are very encouraged that you found our work to be well-written, our method to be clearly presented, and recognized its potential for meaningful impact. We sincerely appreciate the time you have taken to provide this valuable feedback. Your questions have highlighted several areas where we can provide further clarification and strengthen our contribution.
>
> Below, we address the weaknesses and questions you raised.
>
> ---
>
> ### **On Comparison with Traditional Methods: Scalability, Cost, and** **Convergence** **(Weaknesses 2 & 3, Questions 1 & 2)**
>
> We thank you for raising these critical points regarding the comparison with traditional methods. This comparison is central to our work, and we have dedicated **Section 4.4 ("Comparative Experiment with Traditional Algorithms")** and **Table 5** in our paper to a direct discussion and data presentation on this topic.
>
> **The Core Advantage: Overcoming the "Curse of Dimensionality"**
>
> It is essential to distinguish between the challenges of high-dimensional problems. For traditional methods like the Finite Difference Method (FDM), the primary bottleneck is not the algebraic eigenvalue solver itself (e.g., the Power Method or ARPACK), but rather the initial **discretization** **of the continuous operator into a matrix**. This process suffers from the "curse of dimensionality": to maintain accuracy in higher dimensions, the number of grid points must grow exponentially, leading to an intractable explosion in memory and computational cost.
>
> Neural network-based methods and traditional algorithms excel in different domains. In low-dimensional problems, traditional methods are often faster and highly accurate. However, in high-dimensional settings, the cost of discretization makes them impractical, which is precisely where methods like STNet offer a significant advantage.
>
> **Analysis of Computational Cost and Memory Usage**
>
> Our experimental data clearly illustrates this trade-off:
>
> - **The Dilemma of Traditional Methods**: As shown in **Table 5** for the 5D Harmonic problem, for FDM to reduce the relative error of the first eigenvalue (λ1) from $3.20×10^{−1}$ to a modest $3.82×10^{−3}$, the memory consumption skyrockets from 0.0001 GB ($4^5$ grid) to **22.9 GB** ($45^5$ grid). Even with this massive resource usage, the resulting accuracy is orders of magnitude lower than STNet's. **The Efficiency of STNet**: In contrast, STNet achieves a much higher accuracy of $4.62×10^{−5}$ while using only **1.14 GB** of memory.
>
> To further address your question, we provide a more detailed analysis of the convergence time and absolute error of the principal eigenvalue for a five-dimensional Harmonic problem. It is important to note that our traditional baseline in the paper uses `scipy.eigs`, which is based on the highly-optimized ARPACK library—a much more powerful solver than a basic power method.
>
> | Method | Grid/Sample Pts | Memory (GB) | Final Abs. Error | Time to reach error of 1e+0 | Time to reach error of 1e-1 | Time to reach error of 1e-2 | Time to reach error of 1e-3 | Total Time |
> | :--- | :--- | :--- | :--- | :--- | :--- | :--- | :--- | :--- |
> | **FDM+eigs** | 8e+05 (15^5) | 9.1e-02 | 3.2e-02 | 1.3s | 4.0s | 1.1e+01s | - | 1.1e+01s |
> | | 1e+07 (25^5) | 1.2e+00 | 1.2e-02 | 2.5e+01s | 3.2e+01s | 8.1e+01s | - | 1.1e+02s |
> | | 5e+07(35^5) | 6.5e+00 | 6.3e-03 | 1.8e+02s | 2.5e+02s | 4.9e+02s | 1.9e+03s | 1.9e+03s |
> | | 2e+08(45^5) | 2.3e+01 | 3.8e-03 | 9.4e+02s | 3.1e+03s | 6.7e+03s | 1.9e+04s | 1.9e+04s |
> | **STNet** | 5e+04 | 9.5e-01 | 3.4e-04 | 7.2s | 1.4e+01s | 1.7e+02s | 1.3e+03s | 1.0e+05s |
> | | 6e+04 (9^5) | 1.1e+00 | 4.6e-05 | 9.6s | 1.9e+01s | 1.5e+02s | 1.3e+03s | 3.1e+04s |
> | | 7e+04 | 1.3e+00 | 6.5e-05 | 5.7s | 1.1e+01s | 1.4e+02s | 3.0e+02s | 5.4e+04s |
> | | 8e+04 | 1.5e+00 | 2.5e-05 | 8.9s | 1.6e+01s | 1.6e+02s | 9.8e+02s | 2.9e+04s |
>
>
> This detailed data further validates our conclusion. For large-scale (high-dimensional) problems, **STNet holds a significant advantage in both computational cost and memory usage**. Its core strength lies in its **mesh-free random sampling strategy**, which leverages the expressive power of neural networks to approximate eigenfunctions directly. This approach completely bypasses the step of constructing and storing a massive matrix, thereby effectively circumventing the curse of dimensionality that plagues traditional grid-based methods.
>
> The convergence process of all experimental solutions and their computational time and memory requirements will be added to the appendix of the subsequent official version.
>
> ### **On Applicability to Real-World Problems (Question 3)**
>
> This is an excellent question regarding the practical scope of our method. The iterative nature of STNet, which is inspired by the power method, requires the ability to perform an "operator action"—that is, to compute the result of the operator $\mathcal{L}$ acting on the current function approximation (the neural network's output).
>
> The applicability of STNet to a given problem therefore hinges on this requirement:
>
> - If the mathematical form of the operator is known, even if it's highly complex or the domain is irregular, the operator action can be precisely computed using automatic differentiation. In these cases, STNet is directly applicable and its mesh-free nature is a distinct advantage.
> - If the operator is not analytically defined (e.g., it exists as a "black box" simulator) or the data is noisy, our framework could still apply **as long as this operator action can be performed or simulated**. However, if there is no way to evaluate the effect of the operator on a given function, then our method in its current form would not be applicable.
>
> In summary, the ability to compute the operator action is the main prerequisite. Extending the framework to cases where the operator itself must be learned from noisy data (i.e., inverse problems) is a very important direction for future research.
>
> ---
>
> #### **Thanks again**
>
> We sincerely thank you again for your constructive and detailed feedback, which has helped us identify clear pathways to improve our paper. Should you have any further questions or require additional discussion, please don't hesitate to reach out. If we have adequately addressed your concerns, we would be grateful for your consideration in adjusting your evaluation score accordingly.

---

> ### Comment · Reviewer_btcB · 2025-08-06
>
> Thanks for providing detailed data.  I have two questions after reading the reply.
> 1. Is a larger neural network needed for high high-dimensional system? And how does the number of parameters affect the convergence?
> 2. What data type was used in the neural network?

---

> ### Author Response · Authors · 2025-08-06
> **Response to Follow-up Questions from Reviewer btcB (1/2)**
>
> Dear Reviewer,
>
> Thank you for your insightful follow-up questions. We greatly appreciate the opportunity to provide further clarification on these important aspects of our work.
>
> ---
>
> ### **On Network Size, System Dimensionality, and** **Convergence** **(Q1)**
>
> Yes, as you said, a larger neural network is generally required to effectively model higher-dimensional PDE systems, as the complexity of the target eigenfunction increases with the spatial dimension.
>
> Specifically, as detailed in Appendix C.2 (Page 15) of our paper, we configured the STNet model with an increasing number of parameters for higher dimensions, consistent with the setup for the PMNN baseline. Each eigenfunction is approximated by its own MLP, with architectures as follows:
>
> - 1D Problem: [1, 20, 20, 20, 20, 1]
> - 2D Problem: [2, 20, 20, 20, 20, 1]
> - 5D Problem: [5, 40, 40, 40, 40, 1]
>
> Our research has shown that using an under-parameterized model (e.g., applying the 2D architecture to a 5D problem) fails to converge, as the network lacks the capacity to approximate the more complex high-dimensional function.
>
> However, the relationship between parameter count and convergence accuracy is nuanced. Simply increasing the number of parameters for a fixed-dimensional problem does not guarantee better performance and can even be detrimental. To illustrate this, our ablation study on the 2D Harmonic problem (Appendix D, Table 8, Page 16) examined the effect of network width while keeping the depth and number of sample points constant. The absolute errors for the eigenvalues are reported below:
>
> | Eigenvalue | Width 10 | Width 20 | Width 30 | Width 40 |
> | ---------- | -------- | -------- | -------- | -------- |
> | λ₁         | 1.68E-06 | 1.42E-05 | 3.26E-05 | 1.57E-05 |
> | λ₂         | 3.82E-01 | 2.96E-01 | 1.50E+00 | 2.67E+00 |
> | λ₃         | 7.54E-01 | 4.17E-01 | 1.59E+00 | 7.93E+01 |
> | λ₄         | 1.71E-01 | 2.00E+01 | 3.52E+02 | 1.50E+02 |
>
> This data shows that for the 2D problem, increasing the network width (and thus, the parameter count) did not improve accuracy. In fact, our further experiments (Appendix D, Table 9) reveal that the number of sampling points is a more critical factor for convergence. When the number of samples is fixed, excessively increasing the model's parameters can lead to overfitting on the sampled points, which degrades the overall accuracy of the eigenfunction approximation.
>
> In summary, we draw the following conclusions:
>
> 1. **Higher-dimensional PDEs require larger networks** to capture the increased functional complexity. An under-parameterized model will fail to converge.
> 2. For a fixed dimensionality and number of sample points, **indefinitely increasing model parameters leads to** **overfitting** and does not improve performance.
> 3. The optimal approach is to **scale the model's parameters in correspondence with the problem's dimensionality and the number of sample points** to ensure robust performance.
>
> Crucially, because STNet inherits the architectural principles of the power method and PMNN [1], it is exceptionally parameter-efficient. For our 5D PDE experiments, STNet requires only ~5,000 parameters per eigenfunction, whereas leading baselines like NeuralEF [2] and NeuralSVD [3] require ~500,000. This significant difference further underscores the efficiency and superiority of our proposed architecture.
>
> Thank you for raising this important point. We will incorporate this detailed analysis and these key findings into the appendix and the main experimental section of the final manuscript to enhance its clarity.

---

> ### Author Response · Authors · 2025-08-06
> **Response to Follow-up Questions from Reviewer btcB (2/2)**
>
> ### **On Data Types Used in the Neural Network (Q2)**
>
> To ensure numerical precision and consistency across all our experiments, we used a uniform data type setting. Specifically:
>
> - **Neural Network Parameters:** All model parameters in STNet were initialized and stored as `torch.float64`.
> - **Input** **Data:** The input to the network is a list of coordinates for the sampling points within the domain, with each coordinate represented as a `float64`.
> - **Output** **Data:** The network's output, which represents the value of the approximate eigenfunction at each sample point, is also a `torch.float64` tensor.
>
> All related code has been provided in the supplementary materials to ensure full transparency and reproducibility.
>
> Thank you for this question. In the final version of our paper, we will add these specific implementation details, along with illustrative code snippets, to the appendix to further improve clarity.
>
> **References:**
>
> [1] Neural networks based on power method and inverse power method for solving linear eigenvalue problems, Computers & Mathematics with Applications, 2023.
>
> [2] Neuralef: Deconstructing kernels by deep neural networks, ICML 2022.
>
> [3] Operator svd with neural networks via nested low-rank approximation, ICML 2024.
>
> ---
>
> We sincerely thank you once again for your detailed and constructive engagement with our work. We hope these detailed explanations fully address your remaining concerns. Should you have any further questions, please do not hesitate to let us know. We remain available for any further discussion. We are very grateful for your time and guidance, and we would be deeply appreciative if you would consider our clarifications in your final evaluation of our work.

---

> ### Author Response · Authors · 2025-08-08
> **To Reviewer btcB: Thanks and Follow-up Discussion**
>
> Dear Reviewer btcB,
>
> Thank you again for your thoughtful and constructive review.
>
> Following up on our submitted rebuttal, we would be very grateful to hear your thoughts, especially as the discussion period is nearing its end. Your feedback is crucial for us to understand if our responses have adequately addressed your concerns and to help us further improve our paper.
>
> We are happy to provide any further clarifications and look forward to discussing this with you.
>
> Best regards,
>
> Authors

---

### Official Review · Reviewer_Saie · 2025-07-02

**Clarity:** 3
**Significance:** 3
**Originality:** 3
**Rating:** 4
**Confidence:** 3

**Summary:**

The authors  propose STNet based on the power method for estimating eigenvalues of linear operators. From the experimental results, it can be observed that there is a significant improvement in accuracy compared to previous neural network methods and traditional methods. The experimental analysis also roughly leads to the conclusion that STNet's two innovations, Deflation Projection and Filter Transform, can play their respective roles in different scenarios.

**Questions:**

see weaknesses

**Ethical Concerns:**

["NO or VERY MINOR ethics concerns only"]

**Limitations:**

yes

**Quality:**

3

**Strengths And Weaknesses:**

### Strengths:
1. Deflation projection and filter transform are simple and intuitively effective.
2. The paper is well-written with clear and fluent presentation that makes the methodology easy to understand.

### Weaknesses:
Several issues need to be addressed or supplemented:
1. Convergence and stability analysis with graphical illustrations is needed.
2. It is observed that the absolute error of λ in the w/o F and D case is still much lower than other methods when converted to relative error. Does this mean that the performance improvement does not come from F and D? Additionally, consider showing the results of PMNN (only showing principal eigenvalue is OK).

---

> ### Author Rebuttal · Authors · 2025-07-31
>
> Dear Reviewer,
>
> Thank you for your valuable feedback and for recognizing the strengths of our work. We appreciate your positive assessment of our paper's clarity and the intuitive effectiveness of our proposed modules. We have carefully considered your suggestions and provide the following responses, which we hope will address your questions and further strengthen our submission.
>
> ---
>
> ### **On** **Convergence** **and Stability Analysis (Weakness 1)**
>
> We agree that a graphical illustration of convergence is essential for a complete analysis. To provide a preliminary view of the convergence behavior of STNet, we present the absolute error at different iteration stages for the Harmonic problem below. The empty cells in the table indicate that convergence for that particular eigenvalue was already achieved.
>
> Absolute Error vs. Iterations for Harmonic Problem
>
> | Dim  | Eigenvalue Absolute Error | Iter: 100 | 500     | 1,000   | 40,000  | 80,000  | 130,000 | 180,000 |
> | ---- | ------------------------- | ------------- | ------- | ------- | ------- | ------- | ------- | ------- |
> | d=1  | λ1                        | 1.6e-02       | 3.5e-03 | 3.2e-04 | 2.6e-04 | 2.7e-05 | 1.2e-03 |         |
> |      | λ2                        | 1.0e+04       | 2.4e+03 | 2.8e+03 | 8.0e+02 | 9.8e-01 | 8.8e-03 |         |
> | d=2  | λ1                        | 6.9e-02       | 6.9e-03 | 5.9e-03 | 5.3e-06 |         |         |         |
> |      | λ2                        | 6.3e+01       | 5.9e+00 | 4.2e+00 | 5.1e-02 |         |         |         |
> | d=5  | λ1                        | 5.8e-02       | 6.6e-03 | 4.8e-04 | 7.8e-03 | 3.9e-03 | 6.2e-03 | 3.9e-03 |
> |      | λ2                        | 6.3e+03       | 4.1e+02 | 1.5e+02 | 2.9e+00 | 5.5e+00 | 3.8e+00 | 1.3e-01 |
>
> Following your excellent suggestion, we will add comprehensive convergence plots (error vs. iterations) for all algorithms across all experiments in the final camera-ready version of the paper to provide a clearer visual analysis of the convergence dynamics.
>
> ### **On the** **Source** **of Performance Improvement and PMNN Comparison (Weakness 2)**
>
> This is an insightful question, and we thank you for the opportunity to clarify. You are correct in your observation. As stated on page 5, line 105, STNet is designed as an enhanced version of the Power Method Neural Network (PMNN). The PMNN framework itself is capable of achieving very high precision for a single eigenvalue.
>
> Therefore, the **"w/o F and D" case in our ablation study (Table 6) is functionally equivalent to the original PMNN**. The high accuracy for the principal eigenvalue ($ \lambda_1$) in this setting is indeed inherited from this strong baseline.
>
> The primary limitation of PMNN, however, is its inability to compute multiple eigenvalues simultaneously. Our key contribution with STNet is the introduction of the **deflation projection and** **filter** **transform**, which extend the high-precision capabilities of the base method to solve for multiple eigenvalues concurrently and more accurately.
>
> The effectiveness of our modules is clearly demonstrated in Table 6. In the "w/o D" setting, the network correctly finds the principal eigenpair $(v_1, \lambda_1)$ but fails to find the subsequent ones, with the errors for $\lambda_2$ and $\lambda_3$ being extremely large. This shows that the network repeatedly converges to the same dominant eigenfunction. The deflation projection module is what resolves this issue, enabling the network to discover multiple distinct eigenfunctions.
>
> To make this clearer, we will explicitly state in the ablation study section of our final paper that the "w/o F and D" setting serves as a direct comparison to the PMNN baseline, and we will revise the text to better articulate how our contributions build upon it to overcome its limitations.
>
> ---
>
> #### **Thanks again**
>
> We sincerely thank you again for your constructive and detailed feedback, which has helped us identify clear pathways to improve our paper. Should you have any further questions or require additional discussion, please don't hesitate to reach out. If we have adequately addressed your concerns, we would be grateful for your consideration in adjusting your evaluation score accordingly.

---

### Official Review · Reviewer_pBsz · 2025-07-03

**Clarity:** 2
**Significance:** 3
**Originality:** 2
**Rating:** 4
**Confidence:** 2

**Summary:**

This paper introduces STNet, a neural network-based approach to solving linear operator eigenvalue problems. Traditional numerical methods have the curse of dimensionality problem, so deep learning-based methods have recently been suggested. The authors were inspired by classical numerical techniques, specifically deflation projection and filter transform. By excluding eigenvalues and eigenfunctions which are already been computed and amplifying the spectral region of interest, the authors improve the deep learning-based method.

**Questions:**

1) Could you clarify or justify the points raised above?
2) The method is described as being specific to linear operators in the limitations section. Is this a structural limitation of the approach, or just an implementation choice?
3) Are there no comparisons with other more accurate than FDM or specialized traditional methods?
4) Did you compare with FDM combined with deflation or filtering?

**Ethical Concerns:**

["NO or VERY MINOR ethics concerns only"]

**Final Justification:**

I am raising my score to a 4. However, some important aspects—such as theoretical justification, implementation clarity, and experimental detail—remain somewhat underdeveloped in the current submission.

**Limitations:**

Yes

**Paper Formatting Concerns:**

I have carefully checked the manuscript and found no major formatting issues.

**Quality:**

3

**Strengths And Weaknesses:**

* Strengths:
1) A good attempt to improve existing deep learning-based methods by incorporating classical numerical ideas.

* Weakness:
1) Ambiguity in the Abstract

 Although deflation and filter transform are classical numerical techniques, the abstract presents them as if they are the authors’ original ideas. There is no mention in the abstract that these techniques originate from traditional numerical methods.
 Precisely, since they are described only as tools for resolving spectral distribution issues in deep learning, the abstract creates the impression that these are problems unique to neural network methods. This could easily mislead readers into thinking that the techniques themselves are newly developed by the authors. A clear statement is needed so readers can immediately recognize the origin of these methods.

2) Lack of theoretical justification

 While it is a valid idea to incorporate classical methods, there is not enough theoretical analysis to back up this approach. In particular, there is no guarantee that deflation projection and filter transform, when applied in a deep learning context, will preserve the spectral properties of the original operator.

3) Unclear implementation of the differential operator

 The paper only stated that the differential operator 𝐿 is implemented via automatic differentiation, without further details. It is unclear how L is implemented in code, or how deflation and filtering (lines 7–8 in Algorithm 2) are carried out. The description is too vague.

4) Lack of details in the experimental setup

 The experiments do not state the loss functions used in each case explicitly. It makes it difficult to understand or reproduce the experiments.

5) Main contents and Appendix

It seems that deflation projection and filter transform are key ideas of this paper. Therefore, the content currently in Sections 2.3 and 2.4 should be moved from the appendix into the main text and explained in more detail. Additionally, the differences between the baselines in Section 4 are also mainly described in the appendix; these should be elaborated more clearly in the main body to improve the reader’s understanding.

---

> ### Author Rebuttal · Authors · 2025-07-31
>
> Dear Reviewer,
>
> Thank you for your thorough review and insightful comments on our paper. We sincerely appreciate the time and effort you have dedicated to providing this valuable feedback. We have carefully considered each of your points and offer the following clarifications and planned revisions. We hope these responses will fully address your concerns and lead you to reconsider your assessment of our work.
>
> ---
>
> ### **On the Clarity of the Abstract (Weakness 1)**
>
> We agree with your observation. Our intention was not to claim the invention of deflation projection and filter transforms. Our core contribution is to be the **first to successfully integrate these powerful, classical spectral transformation techniques into a neural network framework for solving operator eigenvalue problems**. As you noted, these methods are well-established in traditional numerical analysis, a point we detail in Sections 2.3 and 2.4 of our paper. To prevent any misunderstanding, we will revise the abstract in the final version to explicitly state that our work builds upon these classical numerical methods, clarifying our unique contribution.
>
> ### **On Theoretical Justification  (Weakness 2)**
>
> This is an excellent point, and we appreciate the opportunity to clarify the theoretical underpinnings of our method. The key insight of STNet is that the spectral transformations are applied directly to the original operator $\mathcal{L}$, transforming it into an equivalent but easier-to-solve problem, $\mathcal{L''}$. This process, detailed in Sections 3.2.1 and 3.2.2, does not modify the neural network architecture or its learning mechanism.
>
> Crucially, the transformation preserves the spectral properties in an exact, predictable way. It creates an **equivalent problem**, not an approximation. Therefore, if the neural network minimizes the loss to zero for the transformed problem $\mathcal{L''}$, it provides an exact solution to the original operator eigenvalue problem. The transformation only impacts the convergence path and final precision, not the validity of the solution. Our extensive experimental results, which demonstrate state-of-the-art accuracy, provide strong empirical validation for this theoretical consistency.
>
> ### **On the Implementation of the Differential Operator (Weakness 3)**
>
> Thank you for pointing out the need for more clarity here. The differential operator $\mathcal{L}$ is constructed and applied using **automatic differentiation (AD)**. AD allows for the exact computation of derivatives of any order by decomposing functions into a sequence of elementary operations and applying the chain rule. For instance, to apply an operator like $\frac{d^2}{dx^2} + \frac{d}{dx}$, we use AD to compute the second derivative and the first derivative of the network's output with respect to its input and sum the results.
>
> The specific computations for deflation and filtering (lines 7–8 in Algorithm 2 ) are detailed by the formulas in the main text, specifically **Equation 9** (page 5, line 127) and **Equation 11** (page 6, line 150).
>
> We have also provided the complete, runnable code in the supplementary materials. For example, the operator application and deflation are implemented as follows:
>
> Python
>
> ```Plain
> # Automatic differentiation to compute Laplaciandef compute_grad(self, u, x):
>     u_x = autograd.grad(u.sum(), x, create_graph=True)[0]
>     return u_x
>
> u_xx = Nonefor i in range(self.d):
>     # ... code to compute second partial derivatives and sum them ...
>     u_xixi = self.compute_grad(self.compute_grad(u, x[i]), x[i])
>     u_xx = u_xx + u_xixi if u_xx is not None else u_xixi
>
> # Application of deflation projection operatordef apply_deflation(u_xx, u, max_vecs, max_vals):
>     Lu = -u_xx
>     # Sequentially subtract projections onto solved eigenfunctionsfor i in range(len(max_vecs)):
>         projection = max_vals[i] * torch.matmul(max_vecs[i], torch.matmul(max_vecs[i].transpose(0, 1), u))
>         Lu = Lu - projection
>     return Lu
> ```
>
> We will incorporate these explanations, along with more detailed pseudocode and illustrative code snippets, into the main paper to make our implementation crystal clear. We also plan to release our final code as a user-friendly library to facilitate broader use.
>
> ### **On Details of the Experimental Setup (Weakness 4)**
>
> We apologize for any lack of clarity. The loss function is explicitly defined in our paper. On page 4, Algorithm 2, line 10, the loss for each network is given as: $Loss_{i}^{k} = \frac{1}{N}\sum_{j=1}^{N} [\tilde{v}_i^{k-1}(x_j)-\tilde{u}_i^{k}(x_j)]^{2}$.
>
> This loss formulation remains consistent across all experiments.  The only part that changes is the definition of the operator $\mathcal{L}$ within the computation of $\tilde{u}\_{i}^{k}$. For full transparency, we will add the specific code snippets for implementing the operator $\mathcal{L}$ for each problem (Harmonic, Schrödinger, Fokker-Planck) to the appendix in the final version. For instance, the implementation of Harmonic operator is:
>
> Python
>
> ```Plain
> # Code snippet for Harmonic operator L
> x = self.x
> u = self.forward(x)
> u_xx = Nonefor i in range(self.d):
>     # ... (AD code as shown in previous point)
> Lu = -u_xx # Lu represents the action of the operator L on u
> ```
>
> ### **On the Organization of** **Main** **Content and Appendix (Weakness 5)**
>
> This is a very helpful suggestion. We agree that the background on deflation projection and filter transform, as well as a more detailed comparison of the baseline methods, are crucial for understanding our paper's context and contribution. In the camera-ready version, we will move these essential details from the appendix into the main body of the paper to improve readability and flow.
>
> ### **On the Method's Applicability to Non-linear Operators (Question 2)**
>
> The paper focuses on linear operators primarily because this is the most fundamental and intensively studied area for neural network-based eigenvalue solvers. However, this focus represents an implementation choice rather than a hard structural limitation. In traditional numerical analysis, many non-linear and quadratic eigenvalue problems are tackled by reformulating them into a sequence of linear problems that are solved iteratively.
>
> Following this established paradigm, STNet can, in principle, be extended to solve these more complex classes of problems. We see the current work as a foundational step, and we plan to develop optimized versions of STNet for non-linear, generalized, and other specific eigenvalue problems in our future research.
>
> ### **On Comparisons with Advanced Traditional Methods (Question 3 & 4)**
>
> This is an important question. As we discuss on pages 8-9 (lines 248-254), the primary bottleneck for traditional methods like FDM in high dimensions is the **discretization error**, not the algebraic eigenvalue solver's error. Techniques like deflation or filtering can improve the accuracy of the matrix eigenvalue solve, but they cannot overcome the fundamental error introduced by discretizing a high-dimensional space onto a grid. Reducing this error requires increasing the grid density, which leads to an exponential increase in memory and computational cost, a problem known as the "curse of dimensionality".
>
> To further validate this point, we have run a new comparison for the 5D Harmonic problem against FDM with Richardson extrapolation, a technique that improves accuracy. The results, shown below, confirm that STNet still achieves significantly higher accuracy with comparable memory usage.
>
> | Method                                                 | Grid/Points | Memory (GB) | lambda_1 Error | lambda_2 Error | lambda_3 Error | lambda_4 Error |
> | ------------------------------------------------------ | ----------- | ----------- | -------------- | -------------- | -------------- | -------------- |
> | FDM (Richardson Extrap.)                               | 255         | 3.40E+00    | 8.60E-03       | 2.90E-02       | 2.90E-02       | 2.90E-02       |
> | FDM (with deflation and filtering; Richardson Extrap.) | 255         | 3.40E+00    | 8.60E-03       | 2.90E-02       | 2.90E-02       | 2.90E-02       |
> | FDM (Central Difference)                               | 455         | 2.20E+01    | 3.80E-03       | 1.20E-02       | 1.20E-02       | 1.20E-02       |
> | FDM (with deflation and filtering; Central Difference) | 455         | 2.20E+01    | 3.80E-03       | 1.20E-02       | 1.20E-02       | 1.20E-02       |
> | STNet                                                  | 95          | 1.10E+00    | 4.60E-05       | 1.60E-05       | 1.00E-05       | 2.30E-04       |
>
> This experiment reinforces that neural network-based methods, which rely on random sampling rather than fixed grids, are inherently better suited for high-dimensional problems. We will add this experiment and a more detailed analysis to the appendix in our final version.
>
> ---
>
> #### **Thanks again**
>
> We sincerely thank you again for your constructive and detailed feedback, which has helped us identify clear pathways to improve our paper. Should you have any further questions or require additional discussion, please don't hesitate to reach out. If we have adequately addressed your concerns, we would be grateful for your consideration in adjusting your evaluation score accordingly.

---

> > ### Comment · Reviewer_pBsz · 2025-08-07
> >
> > I thank the authors for their thoughtful and thorough responses. The integration of classical spectral techniques into a neural framework is a promising direction, and the newly added comparisons, particularly against a high-resolution FDM baseline, are appreciated.
> >
> > Based on these clarifications and the authors’ plans for revisions, I am willing to raise my score to a 4. That said, some concerns remain regarding the current version of the paper. In particular, theoretical justification for using spectral transformations in the learning framework is not fully established, and key implementation details (e.g., operator application and deflation steps) are only partially explained in the main text.  While I support the potential of this line of work, I believe that a more polished and self-contained version would make a stronger case. I encourage the authors to incorporate these refinements in the final version.

---

> ### Author Response · Authors · 2025-08-01
> **An error in our previous response**
>
> We sincerely apologize for the typographical error in our response to **Questions 3 & 4**.
>
> The values for "**Grid/Points**" in the table should be "$25^5$, $25^5$, $45^5$, $45^5$, $9^5$".
>
> This was an oversight on our part during the format conversion process. We deeply regret any confusion this may have caused.

---

> ### Author Response · Authors · 2025-08-07
> **Thanks for Your Feedback and Revision Plan**
>
> We greatly appreciate your constructive feedback and your willingness to raise your score! We are pleased to hear that our responses and the new FDM comparison have helped resolve your primary concerns.
>
> We also acknowledge your valuable suggestions for improvement. In the final version, we will focus on enhancing the theoretical justification and clarifying key implementation details to make the paper more self-contained, as you recommended. We will, of course, also incorporate the newly added results. Thank you again for your guidance.

---

### Official Review · Reviewer_z6YH · 2025-07-03

**Clarity:** 3
**Significance:** 2
**Originality:** 2
**Rating:** 5
**Confidence:** 3

**Summary:**

The paper proposes a network based eigen-solver, following several works in the field. The main idea is solving for several eigenfunctions in parallel, while using the deflation technique to decouple the different eigenvalue problems. This way each STNet module is focused on solving the problem within a pre-specified range of eigenvalues. For the algorithm to work properly, one needs a reasonable estimation of the expected eigenvalue range.  They use a filter transform to amplify the eigenvalues in a certain desired range and improve convergence. The results appear to be good, compared to 3 other NN-based methods, achieving better accuracy on several standard eigenvalue problems. Experiments with various variations and an ablation study is carried out. Code of the algorithm is supplied.

**Questions:**

see above

**Ethical Concerns:**

["NO or VERY MINOR ethics concerns only"]

**Final Justification:**

The authors have answered most of my concerns in the rebuttal including running time, architecture, issues on convergennce etc. I expect to put these new material in the final version. Thus I believe the work can be accepted.

**Limitations:**

see above

**Paper Formatting Concerns:**

no concerns

**Quality:**

3

**Strengths And Weaknesses:**

Strengths:

1. The method appears sensible and may be able to compute well middle-range eigenvalues, which are often harder to compute using iterative methods.

2. Experiments show high accuracy, compared to other similar NN-based algorithms.

3. There is transparency, where code is supplied and could be readily used upon publication.


Weaknesses and questions:

1. I did not see details on architecture of the network. Code is of course not a valid way to communicate the architecture. There seem not to be ablation on the architecture.

2. The method may highly depend on having a good approximation of the eigenvalues expected. Not clear what  happens when the range is erronous.

3. What is the complexity of the process? There are some matrix inversions for instance, which are computationally demanding.

4. Running time, compared to other methods. How many iterations are needed on average?

5. Rate of convergence? Although a full theoretical analysis is very hard and beyond the scope. It would good to show at least a plot of convergence: error vs. iterations (or actual time).

6. Grid resolution. Seems to be only one resolution tested.

7. Are there no more recent works on the subject from 2024 and 2025? Please update the related work section.

---

> ### Author Rebuttal · Authors · 2025-07-31
>
> Dear Reviewer,
>
> Thank you for your thoughtful and constructive feedback on our paper. We are grateful for the time you have taken to review our work and for your valuable suggestions, which have helped us to identify areas for improvement. We have carefully considered all your comments and provide the following clarifications, which we hope will address your concerns. We believe these explanations will further strengthen our paper, and we hope you will consider raising your score.
>
> ---
>
> ### **On Network Architecture and Ablation Studies (Weaknesses 1)**
>
> As noted on page 5, line 108 of our paper, the neural network modules employed in our work are all Multi-Layer Perceptrons (MLPs). The core contribution of STNet lies not in a novel network architecture, but in the use of Deflation Projection and Filter Transform to modify the loss function and the optimization process. For this reason, and to ensure a fair comparison with baseline methods, we adopted the standard MLP architecture used by NeuralSVD and PMNN.
>
> To analyze the impact of the architecture, we conducted ablation studies on the depth and width of the MLP, with the results presented in the appendix on page 16 (Tables 7 and 8). We agree that designing a network architecture more specifically suited to this task is a promising direction for future research.
>
> ### **On the Dependency on Initial Eigenvalue Estimates (Weaknesses 2)**
>
> STNet does not critically depend on a precise initial estimation of the eigenvalues. A key feature of our method is that the Filter Transform module dynamically adjusts its filter function based on the approximate eigenvalues computed by the network at each iteration. This adaptive process progressively refines the search and is the primary reason STNet achieves higher accuracy than other algorithms. In all our experiments, we used broad, non-specific initial ranges such as [-10, 10] or [-100, 100], demonstrating the method's robustness.
>
> ### **On Computational Complexity and Runtimes (Weaknesses 3 & 4)**
>
> We appreciate your concern regarding computational complexity. As detailed on page 6, lines 144-151, our method circumvents the high cost of direct matrix inversion. We simulate the inverse power iteration process by reformulating the loss function, thus avoiding explicit and computationally demanding matrix inversions.
>
> While it is true that neural network-based methods can have a larger computational overhead than traditional methods in low-dimensional problems (as mentioned on page 8, line 248), they offer a distinct precision advantage in high-dimensional settings. We provide a runtime comparison for the 5D Harmonic problem below:
>
> | Method                | Computation Time | Principal Eigenvalue Absolute Error |
> | --------------------- | ---------------- | ----------------------------------- |
> | FDM (45^5 points)     | ~1.9e+04 s        | 3.8e-03                            |
> | STNet (partial)       | ~1.3e+03 s        | 1.0e-03                            |
> | STNet (converged)     | ~3.1e+04 s        | 4.6e-05                            |
> | NeuralEF (converged)  | ~1.2e+04 s        | 3.9e-01                            |
> | NeuralSVD (converged) | ~1.2e+04 s        | 2.8e-02                            |
>
> This comparison highlights two points:
>
> 1. STNet achieves a lower error more rapidly and converges to a significantly more accurate solution than NeuralEF and NeuralSVD.
> 2. The final convergence time for STNet is longer. This is an inherent trade-off of the Filter Transform, which iteratively refines the problem to unlock further optimization, thus requiring more iterations to reach its high-precision potential.
>
> We will include detailed runtime records for all experiments in the final version of the paper.
>
> ### **On the Rate of** **Convergence** **(Weaknesses 5)**
>
> We thank you for this suggestion. A full theoretical analysis of the convergence rate is indeed challenging. To provide empirical evidence, we present the convergence process for the Harmonic problem below. The empty cells indicate that convergence for that eigenvalue has already been achieved.
>
> Absolute Error vs. Iterations for Harmonic Problem
>
> | Dim  | Eigenvalue Absolute Error | Iter:  100 | 500     | 1,000   | 40,000  | 80,000  | 130,000 | 180,000 |
> | ---- | ------------------------- | ---------- | ------- | ------- | ------- | ------- | ------- | ------- |
> | d=1  |  λ1                        | 1.6e-02    | 3.5e-03 | 3.2e-04 | 2.6e-04 | 2.7e-05 | 1.2e-03 |         |
> |      | λ2                        | 1.0e+04    | 2.4e+03 | 2.8e+03 | 8.0e+02 | 9.8e-01 | 8.8e-03 |         |
> | d=2  | λ1                        | 6.9e-02    | 6.9e-03 | 5.9e-03 | 5.3e-06 |         |         |         |
> |      | λ2                        | 6.3e+01    | 5.9e+00 | 4.2e+00 | 5.1e-02 |         |         |         |
> | d=5  | λ1                        | 5.8e-02    | 6.6e-03 | 4.8e-04 | 7.8e-03 | 3.9e-03 | 6.2e-03 | 3.9e-03 |
> |      | λ2                        | 6.3e+03    | 4.1e+02 | 1.5e+02 | 2.9e+00 | 5.5e+00 | 3.8e+00 | 1.3e-01 |
>
> Given the space constraints of the rebuttal, we will add comprehensive convergence curves (error vs. iterations and time) for all algorithms in the final manuscript.
>
> ### **On** **Grid** **Resolution Analysis (Weaknesses 6)**
>
> In our main experiments, we fixed the number of sampling points to ensure a fair and direct comparison between methods, with the specific parameters detailed in Appendix C.2 on page 15.
>
> To analyze the effect of this hyperparameter on STNet, we performed a dedicated analysis by varying the number of sampling points, as presented on page 16, line 511 (Table 9). The results in Table 9 show that as long as the number of sampling points is not set too low, STNet maintains its strong performance, demonstrating its robustness to this parameter.
>
> ### **On Updating the Related Work Section (Weaknesses 7)**
>
> Thank you for this valuable suggestion. We have conducted a further literature search, including a review of recent citations for key papers like PMNN and NeuralEF. Our search identified several recent papers that focus on optimizations for specific eigenvalue problem scenarios. However, we did not find new works proposing general-purpose methods for computing differential operator eigenvalues that would constitute direct baselines. We will gladly incorporate these newly found, relevant papers into the related work section in the final version to provide a more current context.
>
> ---
>
> ### **Thanks again**
>
> We sincerely thank you again for your constructive and detailed feedback, which has helped us identify clear pathways to improve our paper. Should you have any further questions or require additional discussion, please don't hesitate to reach out. If we have adequately addressed your concerns, we would be grateful for your consideration in adjusting your evaluation score accordingly.

---

> ### Author Response · Authors · 2025-08-05
> **Thanks for Your Feedback and Revision Plan**
>
> We greatly appreciate your constructive feedback and your willingness to raise your scores! We are pleased to hear that our responses have resolved most of your concerns. As suggested, we will incorporate the newly added results into the final version of the paper.

---

### Decision · Program_Chairs · 2025-09-17

**Decision:**

Accept (poster)

**Comment:**

This paper received the ratings of (5, 4, 4, 4) and the ratings converged toward acceptance after rebuttal. The reviewers recognized STNet's novel integration of classical spectral transformation techniques (deflation projection and filter transform) into neural network-based eigenvalue solvers, achieving state-of-the-art accuracy on benchmark problems. The main concerns centered on limited theoretical justification for why these transformations preserve spectral properties in the neural framework and the lack of real-world applications beyond synthetic benchmarks. The authors provided a strong rebuttal with new convergence analyses, additional experiments comparing against FDM with Richardson extrapolation, and clarified implementation details. All reviewers moved toward acceptance, with reviewer z6YH explicitly supporting acceptance and reviewer pBsz raising their score to borderline accept. While theoretical gaps remain, the empirical evidence and methodological contribution of successfully combining classical numerical techniques with deep learning justify acceptance. The authors should incorporate the promised convergence plots, theoretical clarifications, and additional experimental details in the final version.